# Expressivity Saturation: Reduced Affine Region Usage Under Increasing Task Complexity

**Xuan Qi**[*,†]                                                                                   *xuan.qi@iit.it*
*AI for Good (AIGO), Istituto Italiano di Tecnologia*
*DITEN, University of Genoa*

**Yi Wei**[*]                                                                                   *ywei@smail.nju.edu.cn*
*State Key Laboratory of Novel Software Technology*
*School of Intelligence Science and Technology*
*Nanjing University*

**Fanqi Yu**                                                                                   *fanqi.yu@iit.it*
*AI for Good (AIGO), Istituto Italiano di Tecnologia*
*DITEN, University of Genoa*

**Manuel Lecha**                                                                                   *manuel.lecha@iit.it*
*Istituto Italiano di Tecnologia*

**Reviewed on OpenReview:** *https://openreview.net/forum?id=JiyZE3yKv8*

## Abstract

Piecewise-affine neural networks (e.g., with ReLU or LeakyReLU activations) implement continuous piecewise-affine maps, and the number of affine regions provides a natural proxy for expressive capacity. However, the gap between theoretical region capacity and the affine regions realized after training remains insufficiently understood. We study this gap from two complementary perspectives. First, we give a rigorous, architecture-dependent theorem for affine line-segment probes: for multilayer perceptrons with piecewise-affine activations, the number of affine pieces realized along an affine line-segment probe is upper bounded by an explicit product of layer-wise width terms (and activation breakpoint factors). This yields a neuron-threshold lower bound for representing target functions with prescribed one-dimensional piece complexity, formalizing the minimal region budget required for complex signals. Second, we exactly enumerate affine regions realized within bounded 2D and higher-dimensional domains under controlled task complexity. Under fixed architectures and training protocols, increasing input–label complexity yields trained solutions with markedly fewer realized regions in the evaluation domain, even though worst-case architectural capacity is unchanged; we call this reduced region usage expressivity saturation. Moreover, in the most challenging regimes, 2D visualizations show that region-usage collapse often coincides with degraded decision boundaries. Finally, we visualize the training dynamics of affine-region partitions and decision boundaries, revealing a consistent refinement process during optimization.

## 1 Introduction

Neural networks with piecewise-affine activations (e.g., ReLU Nair & Hinton (2010) and LeakyReLU Maas et al. (2013)) define continuous piecewise-affine (CPWA) maps. This viewpoint provides a concrete and interpretable representation of network expressivity: the input space is partitioned into affine regions on which

---

[*]Equal contribution.
[†]Corresponding author.

the network acts affinely, and the number and geometry of these affine regions quantify how the network allocates functional complexity Montúfar et al. (2014); Li & Wang (2025). During training the affine-region partition evolves over epochs, with successive layers progressively refining earlier partitions and yielding increasingly intricate decision structures Wei et al. (2026b).

A large body of prior work studies the *maximum* number of regions that a given architecture can realize and shows strong depth-dependent gains in worst-case constructions Chen et al. (2023); Hertrich et al. (2023); Averkov et al. (2025). However, these combinatorial capacity results do not directly explain what happens in trained models: in practice, the number of *realized* affine regions is often much smaller than the theoretical maximum, and it depends strongly on initialization, optimization, and data (Tseran & Montúfar, 2021; Milkert et al., 2025). This leaves an important gap between *expressive capacity* and *expressive utilization.*

This gap is especially relevant when the input–label relation is highly complex, i.e., when accurately representing the target may require rapid changes of the decision function at fine input scales. In such regimes, a network may have large worst-case theoretical capacity, yet optimization may still converge to solutions that utilize only a small fraction of that capacity in terms of realized affine partitions. Motivated by this discrepancy, we ask:

> *For a fixed architecture and training protocol, how does increasing complexity of the input–label relation affect the number of affine regions that are actually realized within a bounded domain, and how does this relate to the emergence (or failure) of decision boundaries?*

**Our perspective.** We address this question via a combined theoretical and empirical analysis under fixed architectures and training protocols. On the theoretical side, we analyze one-dimensional probes (line segments in input space), where the affine-piece structure of CPWA networks is exactly countable and yields architecture-dependent bounds. On the empirical side, we perform exact bounded-domain enumeration of realized affine regions in 2D and higher dimensions under controlled task complexity, and we use 2D decision-boundary visualizations and training-time snapshots to connect region formation to classification behavior and its optimization dynamics. The 1D theoretical results are probe-restricted and do not directly imply the higher-dimensional empirical observations; rather, the two parts of the paper are intended to be complementary.

This perspective leads to a central phenomenon that we call *expressivity saturation*: in the controlled low-dimensional fully-connected ReLU MLP settings studied here, under fixed architectures and training protocols, increasing task complexity in our random-label/sample-size regimes can sharply reduce the number of *realized* affine regions exhibited by the trained network within a bounded domain, even though the architecture's worst-case theoretical region capacity is unchanged. In the most extreme regimes we tested, this reduction in realized affine regions coincides with a failure to form effective decision boundaries, indicating a regime where optimization yields both low region usage and poor classification geometry. In this paper, *expressivity saturation* denotes the empirically observed reduction in realized affine-region usage in the controlled low-dimensional fully-connected ReLU MLP regimes studied here, where random labels and increasing sample size are used as a proxy for increasing task complexity.

Our contributions are as follows:

1. **A rigorous 1D affine-region budget theorem.** We prove a deterministic, architecture-dependent upper bound on the number of affine pieces that a piecewise-affine MLP can realize along any affine line-segment probe. As a corollary, we obtain a neuron-threshold lower bound for representing targets with prescribed one-dimensional piece complexity.

2. **Empirical evidence of expressivity saturation in controlled low-dimensional ReLU MLP settings.** By enumerating realized affine regions in 2D and higher-dimensional settings, we empirically demonstrate that, for fixed fully-connected ReLU MLP architectures and training protocols in our controlled random-label/sample-size regimes, increasing task complexity can sharply reduce the number of realized regions within a bounded domain, despite unchanged worst-case theoretical capacity.

3. **Geometric and dynamical evidence linking region collapse to decision-boundary failure.** Our 2D visualizations show that, in the most challenging regimes we tested, an abrupt reduction in realized affine regions often coincides with the failure to form effective decision boundaries, connecting region-usage collapse to poor boundary geometry in practice. Tracking affine partitions and decision boundaries throughout training further reveals a refinement process.

## 2 Related Work

### 2.1 Counting Affine Regions

Understanding the geometric complexity of piecewise-affine neural networks (PANNs) has attracted significant attention. Early efforts include Tseran & Montúfar (2021), who proposed an algorithm to compute the number of affine regions in maxout networks. Berzins (2023) introduced a polyhedral extraction method that operates on subdivided edges rather than explicitly enumerating full regions. Humayun et al. (2023) proposed SplineCam, a technique grounded in PANN theory for high-fidelity computation of network geometry. In the context of graph neural networks, Chen et al. (2023) derived tight upper and lower bounds on the number of affine regions. For convolutional ReLU networks, Xiong et al. (2024) analyzed both the maximal and average number of regions. Leveraging tropical geometry, Piwek et al. (2023) explicitly calculated the number of boundary and affine segments. To mitigate the exponential growth of affine regions with depth, Li & Wang (2025) proposed an accuracy-based approach that reduces this growth to a polynomial function of width.

### 2.2 Evolution of Affine Regions

Beyond counting, the dynamics of affine-region formation have also been studied. Balestriero & Baraniuk (2018) employed a geometric framework to analyze how PANNs hierarchically organize input signals. Cohan et al. (2022) conducted empirical analyses of how the count and density of affine regions evolve in reinforcement learning tasks with continuous control. Grigsby & Lindsey (2022) examined the relationship between ReLU network architecture and the configuration of decision regions. In addition, Balestriero & LeCun (2024) showed that uniform sampling becomes exponentially inefficient for recovering small-volume regions as input dimensionality grows.

### 2.3 Expressive Capacity of Neural Networks

The expressive capacity of PANNs has been investigated from multiple perspectives. The roles of network depth and width have been characterized in (Goujon et al., 2024; Phuong & Lampert, 2020; Hanin et al., 2022; Wang, 2022; Averkov et al., 2025; Hu et al., 2022; Qi et al., 2026), while the influence of training parameters on expressiveness has been examined in (Tiwari & Konidaris, 2022; Rolnick & Kording, 2020; Zhang & Wu, 2020; Trimmel et al., 2021; Milkert et al., 2025). Furthermore, extensions of the universal approximation theorem (Anthony & Bartlett, 2002) have been developed in (He et al., 2020; Hertrich et al., 2023; Chen et al., 2022; Haase et al., 2023; Nguyen et al., 2025), providing theoretical foundations for the representational power of PANNs. Complementary to these global expressivity results, recent work on local complexity in ReLU networks characterizes expressive capacity through the density of affine regions over the input distribution and links it to representation learning and optimization dynamics (Patel & Montúfar, 2025; Wei et al., 2026a).

**Positioning of our contribution.** Our work complements prior studies on worst-case expressivity by linking the complexity of the input–label relation (under controlled setups and fixed training protocols) to the *realized* affine-region usage of a fixed PANN. We derive a line-restricted region upper bound with a corresponding neuron-threshold lower bound, and use *exact* bounded-domain enumeration to show that, in challenging regimes, trained networks may exhibit region-usage collapse together with impaired decision boundaries.

# 3 Preliminaries

## 3.1 Piecewise-Affine Neural Networks and Affine Regions

We consider a fully-connected multilayer perceptron (MLP) equipped with a continuous piecewise-affine (CPWA) activation function. Let $\sigma : \mathbb{R} \to \mathbb{R}$ be a scalar CPWA activation function. By definition, there exists a finite set of breakpoints

$$-\infty = t_0 < t_1 < t_2 < \cdots < t_K < t_{K+1} = \infty$$

such that on each open interval $(t_{k-1}, t_k)$ for $k \in \{1, \ldots, K+1\}$, $\sigma$ acts as an affine transformation

$$\sigma(z) = c_k z + d_k.$$

For instance, the ReLU activation has a single breakpoint at $t_1 = 0$ with $K = 1$, resulting in two affine pieces.

The PANN $f : \mathbb{R}^d \to \mathbb{R}^m$ of depth $L$ is recursively defined as

$$
\begin{aligned}
h_0(x) &= x, \\
h_\ell(x) &= \sigma(W_\ell h_{\ell-1}(x) + b_\ell), \qquad \ell = 1, \ldots, L, \\
f(x) &= W_{L+1} h_L(x) + b_{L+1},
\end{aligned}
\tag{1}
$$

where $x \in \mathbb{R}^d$ is the input, $n_\ell$ denotes the width (number of neurons) of the $\ell$-th hidden layer, $W_\ell \in \mathbb{R}^{n_\ell \times n_{\ell-1}}$ and $b_\ell \in \mathbb{R}^{n_\ell}$ are the weight matrix and bias vector for layer $\ell$, and the activation $\sigma$ is applied element-wise.

The piecewise-affine nature of $\sigma$ induces a partition of the input space $\mathbb{R}^d$ into distinct regions based on the activation states of the hidden neurons. Let $\theta$ represent the fixed network parameters.

**Definition 1** (Affine region Montúfar et al. (2014)). Fix the network parameters $\theta$. An *affine region* of the network $f(\cdot; \theta)$ is a maximal connected subset $\Omega \subset \mathbb{R}^d$ with non-empty interior on which the mapping $x \mapsto f(x; \theta)$ is affine.

For deep networks with CPWA activations, these affine regions are determined by the intersection of pre-activation conditions across all layers and form polyhedral cells. To obtain a mathematically exact and tractable notion of one-dimensional complexity, we next restrict attention to *affine line-segment probes*.

## 3.2 Line-Restricted Affine Complexity

To rigorously study realized expressivity in one dimension, we restrict the network to affine line segments in input space.

**Definition 2** (Affine line-segment probe). An *affine line-segment probe* is a map $\gamma : [0,1] \to \mathbb{R}^d$ of the form

$$\gamma(s) = x_0 + s\,v, \qquad s \in [0,1], \tag{2}$$

for some $x_0, v \in \mathbb{R}^d$ with $v \neq 0$.

For such probes, the composition $f \circ \gamma$ is a CPWA function of the scalar parameter $s$, so its affine-piece structure is well defined and exactly countable.

**Definition 3** (Line-restricted region count). Let $\gamma : [0,1] \to \mathbb{R}^d$ be an affine line-segment probe. The *line-restricted affine region count* of a network $f$ along $\gamma$ is the number of affine pieces formed by the composition $f \circ \gamma$. Formally,

$$\alpha(f; \gamma) := \min \left\{ R \in \mathbb{N} \;\middle|\; \begin{array}{l} \exists \text{ a partition } 0 = \tau_0 < \tau_1 < \cdots < \tau_R = 1 \text{ such that} \\ f \circ \gamma \text{ is affine on each open interval } (\tau_{r-1}, \tau_r) \end{array} \right\}. \tag{3}$$

This quantity is exact, mathematically tractable, and directly captures the number of affine transformations the network deploys along an affine line probe in input space.

### 3.3 Target Complexity Along a Probe

To mathematically ground our investigation into the minimum network capacity required to express highly irregular one-dimensional signals, we define the intrinsic affine complexity of a target map on the probe parameter domain.

**Definition 4** (1D piece complexity). Let $g : [0,1] \to \mathbb{R}^m$ be a target mapping. Its *one-dimensional piece complexity* is defined as

$$Q(g) := \min \left\{ M \in \mathbb{N} \; \middle| \; \begin{array}{l} \exists \text{ a partition of } [0,1] \text{ into } M \text{ connected intervals} \\ \text{on each of which } g \text{ is affine} \end{array} \right\}. \tag{4}$$

If $g$ cannot be represented as a finite piecewise-affine function on $[0,1]$, we define $Q(g) = \infty$. For classification tasks restricted to a probe, $g$ can be instantiated as the classification margin; consequently, the number of distinct label alternations along the probe parameter provides a lower bound on $Q(g)$. This metric serves as the complexity threshold in our capacity analysis.

## 4 A Rigorous Capacity Bound for CPWA Networks on 1D Probes

### 4.1 Architecture-dependent upper bound on line regions

We consider CPWA activation functions with $K$ breakpoints. Along any affine line-segment probe, each neuron receives an affine scalar pre-activation on every incoming affine piece, and therefore can introduce at most $K$ split points on that piece.

**Theorem 5** (Line-region upper bound for CPWA MLPs). *Let $f : \mathbb{R}^d \to \mathbb{R}^m$ be a fully-connected neural network with hidden layer widths $(n_1, \ldots, n_L)$, equipped with a CPWA activation function $\sigma$ having $K$ breakpoints (e.g., $K = 1$ for ReLU or LeakyReLU). Let $\gamma : [0,1] \to \mathbb{R}^d$ be an affine line-segment probe. Then the line-restricted region count is upper bounded by*

$$\alpha(f; \gamma) \leq \prod_{\ell=1}^{L} (K n_\ell + 1). \tag{5}$$

**Remark 6.** This is a deterministic upper bound that depends only on the architecture (depth, widths, and activation breakpoint count), independent of initialization or training dynamics. The theorem is *probe-restricted*: it applies to affine line-segment probes and does not, in general, extend to arbitrary nonlinear curves in input space.

### 4.2 Minimum neuron budget for representing a 1D target

**Corollary 7** (Neuron-threshold lower bound on a fixed-depth CPWA architecture). *Let $g : [0,1] \to \mathbb{R}^m$ be a target map with $Q(g)$ affine pieces. Suppose a CPWA network $f$ (with $K$-breakpoint activations) exactly realizes $g$ along an affine line-segment probe $\gamma$, namely*

$$f(\gamma(s)) = g(s), \qquad \forall s \in [0,1]. \tag{6}$$

*Then the network's architectural capacity must satisfy*

$$\prod_{\ell=1}^{L} (K n_\ell + 1) \geq Q(g). \tag{7}$$

*Consequently, by the AM–GM inequality, if the total number of hidden neurons is $N = \sum_{\ell=1}^{L} n_\ell$, then*

$$N \geq \frac{L}{K} \left( Q(g)^{1/L} - 1 \right). \tag{8}$$

*For architectures with equal-width hidden layers ($n_\ell = w$ for all $\ell$), this simplifies to*

$$w \geq \left\lceil \frac{Q(g)^{1/L} - 1}{K} \right\rceil. \tag{9}$$

Corollary 7 states that to express a sufficiently complex one-dimensional target along an affine line probe, the architecture must supply an adequate line-region budget. The complexity burden is mitigated linearly by the activation breakpoint count $K$ and exponentially by the depth $L$.

### 4.3 Practical region-yield heuristic

While the deterministic theorem establishes a strict *capacity* upper bound, practical optimization typically yields significantly fewer regions. Empirically and theoretically, the average number of regions realized along affine line probes often scales roughly linearly with the total number of neurons under standard initializations for PANNs Hanin & Rolnick (2019). This motivates the following practical efficiency metrics.

**Definition 8** (Realized region efficiency on affine line probes)**.** Let $\Gamma$ be a distribution over affine line-segment probes, and let $f$ be a trained CPWA network. We define the mean realized line-region count as

$$\bar{R}(f;\Gamma) := \mathbb{E}_{\gamma \sim \Gamma}[\alpha(f;\gamma)]. \tag{10}$$

The realized region efficiency relative to the theoretical upper bound is defined as

$$\eta_{\mathrm{norm}}(f;\Gamma) := \frac{\bar{R}(f;\Gamma)}{\prod_{\ell=1}^{L}(Kn_\ell + 1)}. \tag{11}$$

Equivalently, it is often useful to measure the *regions-per-neuron* statistic

$$\alpha(f;\Gamma) := \frac{\bar{R}(f;\Gamma)}{N}. \tag{12}$$

In standard training regimes, $\alpha$ often remains within a modest, architecture-dependent range. However, our empirical findings indicate that $\alpha$ can collapse when the intrinsic complexity of the input–label mapping exceeds the effective capacity of the fixed architecture under the given training protocol.

**Practical threshold estimator.** If a target distribution intrinsically requires approximately $Q$ affine pieces along probes sampled from $\Gamma$, we propose the rule of thumb

$$N_{\mathrm{min}}^{\mathrm{practical}} \approx \left\lceil \frac{Q}{\alpha(f;\Gamma)} \right\rceil, \tag{13}$$

where $\alpha(f;\Gamma)$ is estimated empirically for the architecture family, initialization scheme, and optimization algorithm of interest.

**Proofs.** Proofs of the main theoretical results are provided in Appendix A.

## 5 Counting Methodology

We describe how we *exactly enumerate* the number of affine regions realized by a trained PANN within a bounded input domain. Our methodology separates (i) *exact* one-dimensional (1D) counting along affine line probes, used to support the theoretical analysis, from (ii) *exact bounded-domain enumeration* in two-dimensional (2D) and higher-dimensional settings, where we exhaustively traverse *all* affine regions intersecting the prescribed domain $\Omega$.

### 5.1 Exact counting along one-dimensional probes

Let $f : \mathbb{R}^d \to \mathbb{R}^m$ be a multilayer perceptron with piecewise-affine activations, and let $\gamma : [a, b] \to \mathbb{R}^d$ be an affine line segment. Equivalently, by an affine reparameterization of the scalar variable, one may work on a general interval $[a, b]$ instead of $[0, 1]$. Then the composed map $s \mapsto f(\gamma(s))$ is CPWA in the scalar parameter $s$. We compute the exact number of affine pieces by propagating an interval partition of $[a, b]$ through the network layer by layer. On any current interval, each neuron's pre-activation is affine in $s$, and hence can hit each activation breakpoint at most once on that interval. Therefore, for a CPWA activation with breakpoints $\{t_1, \ldots, t_K\}$, each neuron contributes at most $K$ split points per incoming affine interval; for ReLU or LeakyReLU ($K = 1$, breakpoint at 0), this reduces to splitting at pre-activation roots.

---

**Algorithm 1** Exact affine-piece counting along an affine 1D line probe for CPWA MLPs

---

**Require:** Network $f$ with $L$ hidden layers; affine line-segment probe $\gamma : [a, b] \to \mathbb{R}^d$; activation breakpoints $\{t_1, \ldots, t_K\}$

1: Initialize partition $\mathcal{P} \leftarrow \{[a, b]\}$
2: **for** $\ell = 1, \ldots, L$ **do**
3:      $\mathcal{P}_{\text{new}} \leftarrow \emptyset$
4:      **for** each interval $I \in \mathcal{P}$ **do**
5:          Represent $h_{\ell-1}(\gamma(s))$ as an affine function of $s$ on $I$
6:          Compute all activation-breakpoint hits on $I$:
$$\mathcal{Z} \leftarrow \bigcup_{j=1}^{n_\ell} \bigcup_{k=1}^{K} \{ s \in \text{int}(I) : a_{\ell,j}(s) = t_k \}$$
7:          Split $I$ at the sorted points in $\mathcal{Z}$ and add the resulting subintervals to $\mathcal{P}_{\text{new}}$
8:      **end for**
9:      $\mathcal{P} \leftarrow \mathcal{P}_{\text{new}}$
10: **end for**
11: **return** $|\mathcal{P}|$                        ▷ Exact number of affine pieces of $f \circ \gamma$

---

## 5.2 Exact bounded-domain region enumeration in 2D and higher dimensions

Exact global counting of affine regions over $\mathbb{R}^d$ quickly becomes computationally prohibitive as depth, width, and input dimension increase. For $d \geq 2$, we therefore perform *exact bounded-domain enumeration*: we exhaustively enumerate all realized affine cells intersecting a prescribed compact domain $\Omega \subset \mathbb{R}^d$, rather than attempting global enumeration over $\mathbb{R}^d$.

**Bounded-domain objective.** Let $\Omega \subset \mathbb{R}^d$ be a compact domain of interest (e.g., $\Omega = [-1, 1]^d$). We aim to exactly enumerate

$$\mathcal{R}_\Omega(f) := \{\text{affine cells of } f \text{ intersecting } \Omega\},$$

and compute statistics derived from $\mathcal{R}_\Omega(f)$ within $\Omega$ (in particular, the exact region count $|\mathcal{R}_\Omega(f)|$).

**Activation-region representation.** For piecewise-affine MLPs, fixing an activation pattern induces a local affine map $f(x) = Ax + b$ on an associated affine region (possibly one connected component among multiple components sharing the same activation signature). Hence, bounded-domain enumeration can be formulated as traversal over activation-induced candidate regions under feasibility and intersection constraints with $\Omega$.

**Enumeration workflow.** Starting from one or more feasible seeds intersecting $\Omega$, we iteratively (i) recover local affine/hyperplane descriptors for the current region, (ii) identify valid neighboring regions via boundary-crossing operations, (iii) test intersection with $\Omega$, and (iv) register previously unseen regions. When the frontier is exhausted, this procedure yields an *exactly enumerated* region set $\widehat{\mathcal{R}}_\Omega = \mathcal{R}_\Omega(f)$ within $\Omega$, from which region statistics (including region counts) are computed exactly.

# 6 Experiments

## 6.1 Implementation and common setup

All experiments are implemented in PyTorch using fully-connected ReLU MLPs. Hidden architectures are denoted by $[n_1, n_2, \ldots, n_L]$, where each hidden layer is followed by a ReLU and the output layer is linear. Unless otherwise stated, we train with Adam, batch size 32, and cross-entropy loss (classification). For 2D experiments, inputs are normalized to $\Omega = [-1, 1]^2$. In 1D, we compute $\alpha(f; \gamma)$ exactly via interval-partition propagation (Algorithm 1). In 2D, we exactly enumerate all realized affine regions intersecting $\Omega$ by exhaustive bounded-domain region exploration (Algorithm 2). Additional ablations on residual connections under complex

---

**Algorithm 2** Exact bounded-domain affine-region enumeration for $d \geq 2$

---

**Require:** Trained PANN $f$, bounded domain $\Omega \subset \mathbb{R}^d$
**Ensure:** Exactly enumerated region set $\widehat{\mathcal{R}}_\Omega = \mathcal{R}_\Omega(f)$ and exact region statistics in $\Omega$
 1: Find an initial feasible region $r_0$ such that $r_0 \cap \Omega \neq \emptyset$
 2: $\mathcal{Q} \leftarrow \{r_0\}, \quad \widehat{\mathcal{R}}_\Omega \leftarrow \emptyset$
 3: **while** $\mathcal{Q} \neq \emptyset$ **do**
 4:     Pop one candidate region $r$ from $\mathcal{Q}$
 5:     **if** $r \in \widehat{\mathcal{R}}_\Omega$ **then**
 6:         **continue**
 7:     **end if**
 8:     **if** $r \cap \Omega = \emptyset$ **then**
 9:         **continue**
10:     **end if**
11:     Add $r$ to $\widehat{\mathcal{R}}_\Omega$
12:     Compute local descriptors of $r$ (activation pattern / affine form / boundary hyperplanes)
13:     Generate neighboring candidates $\mathcal{N}(r)$ by valid boundary-crossing operations
14:     **for** each $r' \in \mathcal{N}(r)$ **do**
15:         **if** $r' \notin \widehat{\mathcal{R}}_\Omega$ and $r'$ is feasible **then**
16:             Push $r'$ into $\mathcal{Q}$
17:         **end if**
18:     **end for**
19: **end while**
20: **return** $\widehat{\mathcal{R}}_\Omega$ and statistics derived from $\widehat{\mathcal{R}}_\Omega$

---

2D inputs are reported in Appendix B, together with further visualizations of decision-boundary dynamics throughout training in Appendix C.

## 6.2 Exact 1D enumeration

This section uses *exact* 1D affine-piece enumeration to (i) sanity-check our counting pipeline against the proven line-region bound in Theorem 5, (ii) isolate architecture-induced line complexity via an initialization-only stress test (no optimization), (iii) empirically instantiate the neuron-threshold implication in Corollary 7 on controlled targets with known piece complexity, and (iv) quantify the practical region-yield statistics $\bar{R}$, $\alpha$, and $\eta_{\text{norm}}$ that motivate the heuristic discussion in Section 4. All 1D experiments use ReLU activations (i.e., $K = 1$), so the bound in Theorem 5 specializes to $\alpha(f; \gamma) \leq Q_{\text{upper}}$, where $Q_{\text{upper}} = \prod_{\ell=1}^{L}(n_\ell + 1)$.

### 6.2.1 Protocol and metrics

**Exact counting along probes.** For each network $f$ and probe $\gamma$, we compute $\alpha(f; \gamma)$ *exactly* using Algorithm 1. For a probe distribution $\Gamma$, we report the mean region count $\bar{R} = \mathbb{E}_{\gamma \sim \Gamma}[\alpha(f; \gamma)]$, the regions-per-neuron statistic $\alpha = \bar{R}/N$, and the normalized efficiency $\eta_{\text{norm}} = \bar{R}/Q_{\text{upper}}$, where $N = \sum_{\ell=1}^{L} n_\ell$ is the total number of hidden neurons and $Q_{\text{upper}} = \prod_{\ell=1}^{L}(n_\ell + 1)$ for ReLU networks.

**Architectures and probes.** We evaluate three ReLU MLP families, $[32]^5$ ($N = 160$), $[64]^3$ ($N = 192$), and $[128]^3$ ($N = 384$), aggregated over multiple random seeds. Unless otherwise stated, each run uses 200 random line probes for estimating expectations over $\Gamma$.

**Initialization-only stress test.** To remove optimization effects and probe architecture-induced line complexity directly, we additionally conduct an initialization-only stress test with no training updates. For each architecture, we sample $S = 3$ random seeds and $M = 100$ probes per seed, compute $\alpha(f; \gamma)$ exactly, and record the extreme statistic $\text{MaxLR} = \max_\gamma \alpha(f; \gamma)$ as well as the normalized ratio $\rho_{\max} = \max_\gamma \alpha(f; \gamma)/Q_{\text{upper}}$; the resulting trends and ratios are summarized in Figure 1.

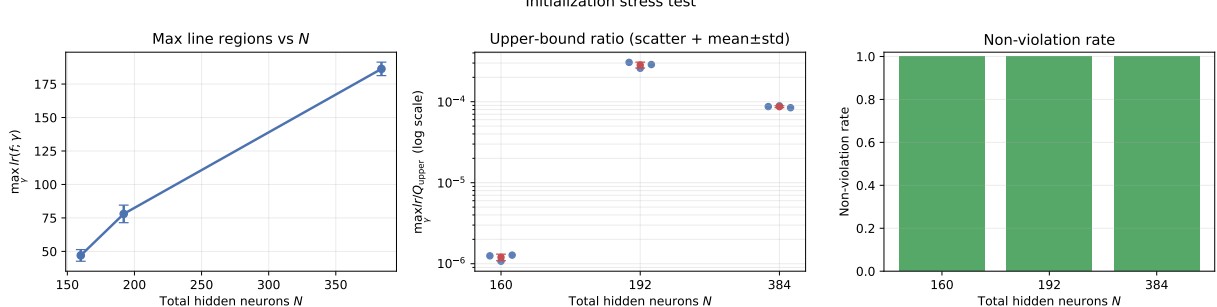

Figure 1: **Initialization-only stress test (exact 1D counting).** Left: $\max_\gamma \alpha(f; \gamma)$ versus $N$ (mean±std across seeds). Middle: per-seed $\rho_{\max}$ scatter with mean±std error bars (log-scale y-axis). Right: non-violation rate of the theoretical upper bound.

**Controlled synthetic threshold suite.** To instantiate Corollary 7 under known target complexity, we train depth-$L = 3$ ReLU MLPs with equal-width hidden layers $[w, w, w]$ (width sweep $w \in \{2, 4, \dots, 64\}$, hence $N = 3w$) on synthetic 1D piecewise-affine targets $g : [0, 1] \to \mathbb{R}$ with piece complexity $Q(g) \in \{4, 8, 12, 16, 24, 32\}$. Each $(Q, w)$ is trained with 3 seeds; a run is deemed successful if the max absolute error is $\leq 2 \times 10^{-2}$ on a dense evaluation grid over $[0, 1]$. We report $\hat{N}_{\min}^{\mathrm{obs}}$ as the smallest $N$ achieving the target success criterion, using both a "loose" rule (success rate $\geq 2/3$) and a "strict" rule (all seeds successful), summarized in Table 1.

### 6.2.2 Consistency checks for Theorem 5

**Trained networks.** Across all aggregated trained runs and probes, we observe no violations of the specialized bound, i.e., `violation_count`=0. The distribution of $\alpha(f; \gamma)$ computed by exact enumeration is shown in Figure 2 (top), and lies far below $Q_{\mathrm{upper}}$, illustrating the gap between worst-case architectural capacity and realized region usage after training. The same figure also reports layer-wise split statistics (bottom), visualizing how affine-piece refinements accumulate through depth.

**Initialization-only stress test.** We also observe no upper-bound violations under random initialization, and MaxLR increases with architecture size; Figure 1 (left) reports $\max_\gamma \alpha(f; \gamma)$ as a function of $N$ (mean±std across seeds). Despite this growth, the normalized extreme ratio $\rho_{\max}$ remains far below 1; Figure 1 (middle) shows per-seed $\rho_{\max}$ values (with mean±std) on a log scale, quantifying substantial slack between realized and worst-case theoretical capacity at random initialization. Finally, Figure 1 (right) reports the non-violation rate, which is 1.0 in all tested settings.

### 6.2.3 Instantiation of Corollary 7 on known-$Q$ targets

For the synthetic suite with $L = 3$ and ReLU, Corollary 7 yields the necessary condition $(w + 1)^3 \geq Q(g)$, equivalently $N \geq L(Q(g)^{1/L} - 1)$. In our experiments, no successful fit contradicts this implication over the tested range (Table 1). This should be read as an empirical *non-refutation* of the necessary condition under our optimization protocol and error criterion, rather than as a sharp characterization of the minimal achievable $N$, since finite optimization budgets can require substantially more neurons than the existence-based lower bound. The observed thresholds in Table 1 are much larger than the theoretical necessary condition, directly quantifying the optimization-limited gap between architectural capacity and realized expressivity. Together with the initialization-only stress test in Figure 1, which already exhibits substantial slack relative to $Q_{\mathrm{upper}}$, these results underscore that worst-case region-capacity bounds can be loose in typical parameter regimes and motivate reporting $\alpha$ as a practical summary statistic for realized region yield.

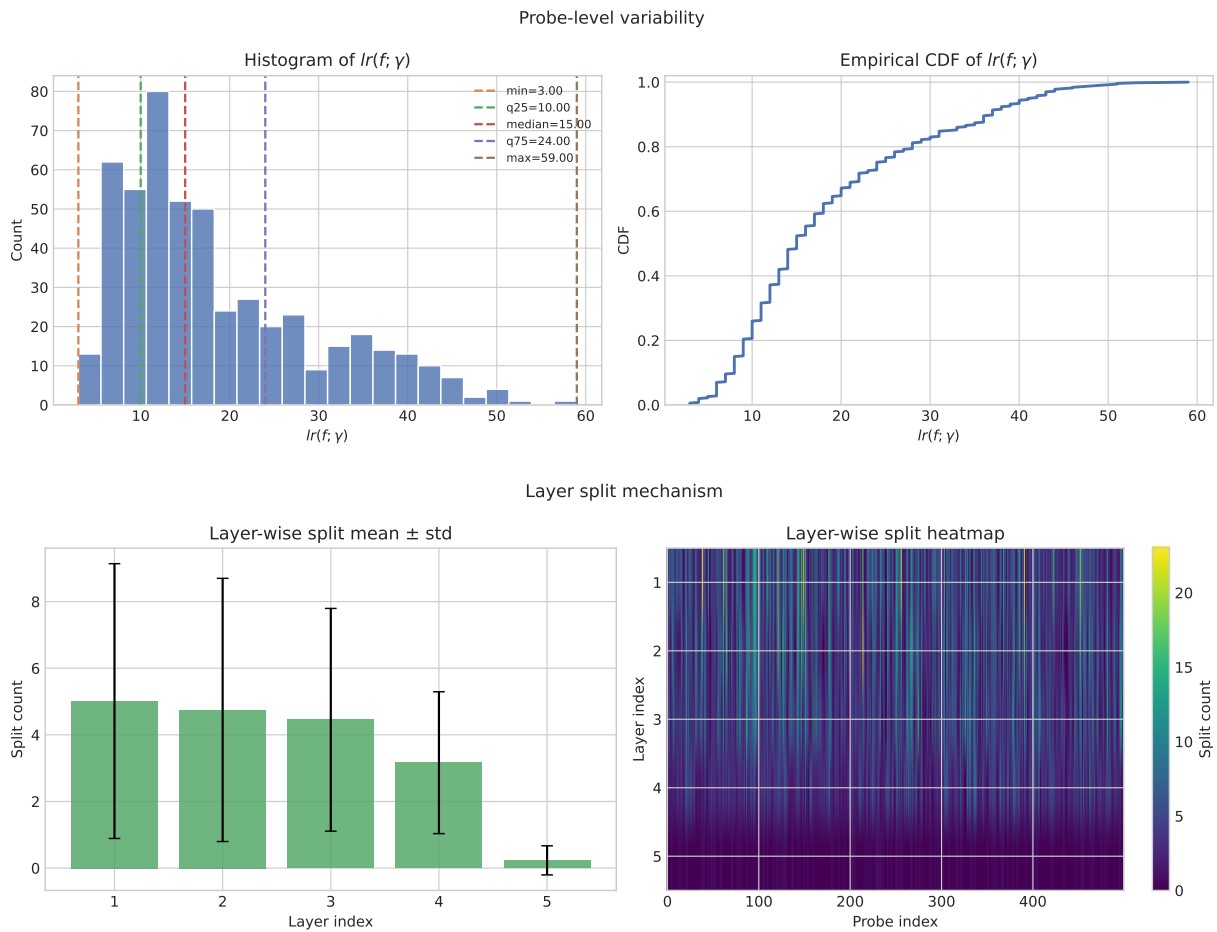

Figure 2: **Exact 1D probe statistics (trained networks).** Top: distribution/CDF of $\alpha(f; \gamma)$ computed by exact enumeration. Bottom: layer-wise split statistics along probes, illustrating how affine-piece refinements accumulate through depth.

Table 1: Synthetic threshold suite: observed neuron budgets versus the necessary-condition lower bound in Corollary 7 ($L = 3, K = 1$).

| $Q(g)$ | $\lceil L(Q^{1/L} - 1) \rceil$ | $\hat{N}_{\min}^{\text{obs}}$ (loose) | $\hat{N}_{\min}^{\text{obs}}$ (strict) | violation |
|:---:|:---:|:---:|:---:|:---:|
| 4 | 2 | 108 | 108 | no |
| 8 | 3 | 150 | 150 | no |
| 12 | 4 | 138 | 138 | no |
| 16 | 5 | 144 | 144 | no |
| 24 | 6 | – | – | no observed success |
| 32 | 7 | – | – | no observed success |

### 6.2.4 Practical region-yield statistics and the $Q/\alpha$ rule

Using the empirically estimated $\alpha$, we evaluate the rule-of-thumb predictor $N_{\min}^{\text{practical}} \approx \lceil Q/\alpha \rceil$. The architecture-level trends of $\bar{R}$, $\alpha$, and $\eta_{\text{norm}}$ are summarized in Figure 3 (top), while Figure 3 (bottom)

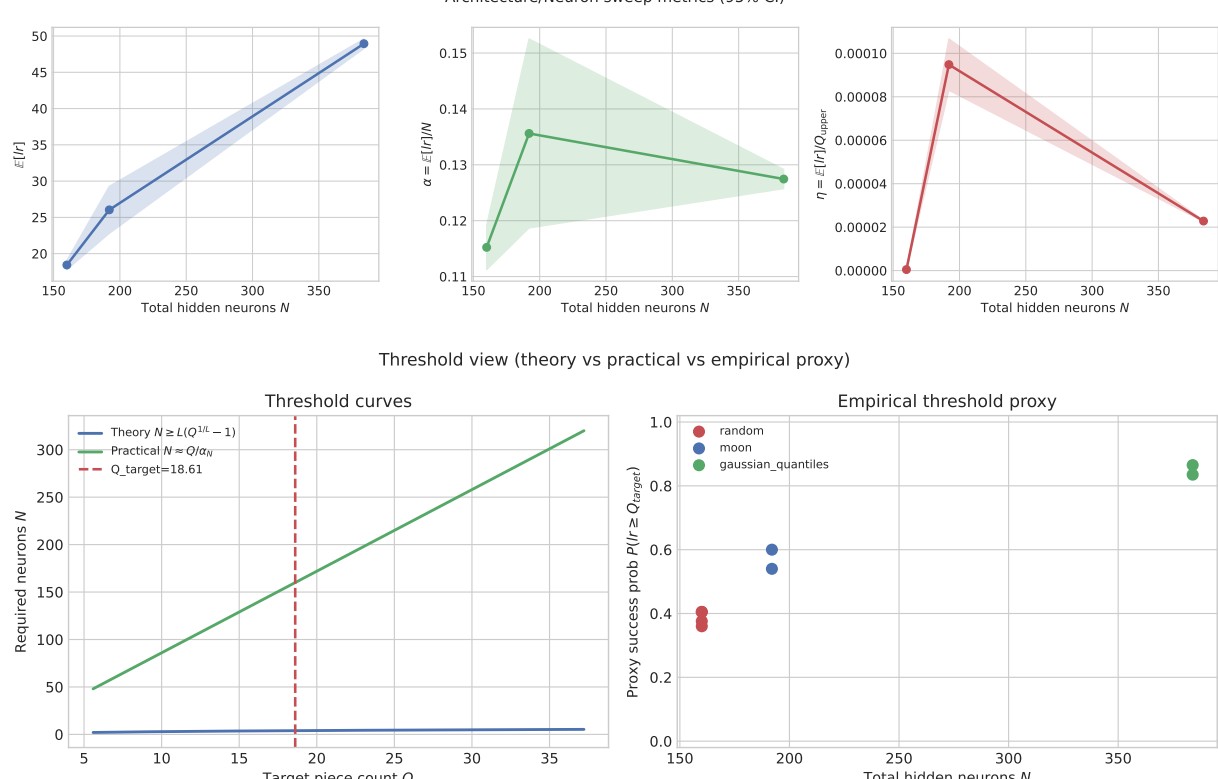

Figure 3: **Architecture-level 1D trends and threshold proxy.** Top: $\bar{R}$, $\alpha$, and $\eta_{\mathrm{norm}}$ versus $N$. Bottom: theoretical/practical curves together with an empirical proxy $P(\alpha \geq Q_{\mathrm{target}})$ from probe statistics.

compares the theoretical necessary-condition curve from Corollary 7 with the heuristic $Q/\alpha$ and an empirical proxy derived from probe statistics.

**Summary (1D).** Exact enumeration provides (i) a broad consistency check of our counting pipeline against the proven ReLU-specialized bound in Theorem 5, including an initialization-only stress test that removes optimization effects (Figure 1), (ii) an empirical non-refutation of the necessary-condition implication in Corollary 7 on controlled targets (Table 1), and (iii) quantitative region-yield statistics $\bar{R}$, $\alpha$, and $\eta_{\mathrm{norm}}$ that summarize the realized-region gap and support the practical discussion in Section 4 (Figures 2 and 3).

### 6.3 2D exact region enumeration and decision-boundary visualization

This section studies realized affine-region formation directly in the input plane and links region usage to classification behavior. All 2D region counts reported here are obtained by *exact enumeration* of all affine regions intersecting the bounded domain $\Omega = [-1, 1]^2$ using Algorithm 2 (bounded-domain exhaustive exploration). We pair exact region counts with decision-boundary visualizations to connect geometric expressivity to classification outcomes.

#### 6.3.1 Expressivity saturation under increasing task complexity

We study random 2D inputs with random labels while increasing the number of training samples, using a fixed ReLU MLP architecture $[32, 32, 32]$ and counting domain $\Omega = [-1, 1]^2$. To quantify variability, all scalar summaries in this subsection are aggregated over 5 random seeds and reported as mean $\pm$ standard deviation. Exact affine-region counts across training epochs and sample sizes are summarized in Table 2. In

Table 2: **Exact** 2D affine-region counts $|\mathcal{R}_\Omega(f)|$ in $\Omega = [-1,1]^2$ for network $[32, 32, 32]$ on random labels, reported as mean $\pm$ standard deviation over 5 random seeds at selected training epochs.

| Epoch \\ Samples | 0 | 10 | 30 | 50 | 80 | 100 | 300 | 500 | 800 | 1000 | 3000 | 5000 | 10000 | 30000 | 50000 |
|---|---|---|---|---|---|---|---|---|---|---|---|---|---|---|---|
| 200 | $1151_{\pm1}$ | $938_{\pm42}$ | $919_{\pm9}$ | $1047_{\pm31}$ | $968_{\pm24}$ | $1219_{\pm46}$ | $1388_{\pm13}$ | $1531_{\pm37}$ | $1821_{\pm28}$ | $1784_{\pm5}$ | $1861_{\pm41}$ | $2139_{\pm19}$ | $1950_{\pm34}$ | $2187_{\pm12}$ | $2436_{\pm27}$ |
| 500 | $1151_{\pm0}$ | $949_{\pm29}$ | $1437_{\pm44}$ | $1402_{\pm11}$ | $1605_{\pm36}$ | $1511_{\pm22}$ | $2109_{\pm47}$ | $2271_{\pm15}$ | $2578_{\pm33}$ | $2730_{\pm6}$ | $3164_{\pm25}$ | $3588_{\pm39}$ | $3929_{\pm14}$ | $4341_{\pm48}$ | $4229_{\pm21}$ |
| 1000 | $1153_{\pm2}$ | $1030_{\pm7}$ | $891_{\pm35}$ | $1021_{\pm18}$ | $1146_{\pm43}$ | $1288_{\pm12}$ | $1661_{\pm30}$ | $2082_{\pm4}$ | $2191_{\pm49}$ | $2233_{\pm23}$ | $2412_{\pm16}$ | $2709_{\pm38}$ | $2774_{\pm10}$ | $3047_{\pm45}$ | $3152_{\pm27}$ |
| 2000 | $1154_{\pm3}$ | $824_{\pm40}$ | $1186_{\pm6}$ | $1349_{\pm32}$ | $1298_{\pm25}$ | $1457_{\pm47}$ | $1218_{\pm9}$ | $1244_{\pm36}$ | $1301_{\pm20}$ | $1360_{\pm3}$ | $1394_{\pm44}$ | $1556_{\pm17}$ | $1502_{\pm28}$ | $1946_{\pm11}$ | $1925_{\pm41}$ |
| 5000 | $1151_{\pm0}$ | $796_{\pm5}$ | $1057_{\pm39}$ | $523_{\pm12}$ | $427_{\pm34}$ | $289_{\pm8}$ | $196_{\pm46}$ | $74_{\pm19}$ | $123_{\pm27}$ | $97_{\pm43}$ | $414_{\pm10}$ | $301_{\pm31}$ | $452_{\pm7}$ | $406_{\pm48}$ | $621_{\pm24}$ |
| 10000 | $1152_{\pm1}$ | $838_{\pm16}$ | $392_{\pm45}$ | $274_{\pm9}$ | $234_{\pm29}$ | $41_{\pm4}$ | $105_{\pm18}$ | $63_{\pm4}$ | $94_{\pm4}$ | $44_{\pm12}$ | $77_{\pm8}$ | $49_{\pm6}$ | $68_{\pm6}$ | $46_{\pm7}$ | $41_{\pm10}$ |

Table 3: **Exact** region counts at epoch 10000 for random labels with 5000 samples under initialization/optimizer ablations (network $[32, 32, 32]$, domain $\Omega = [-1,1]^2$).

| Configuration | Exact #regions in $\Omega$ at epoch 10000 |
|---|---|
| Baseline setting (as in Table 2) | $452_{\pm7}$ |
| Xavier initialization (other settings unchanged) | $579_{\pm12}$ |
| SGD optimizer (other settings unchanged) | $962_{\pm6}$ |

addition, Figure 4 reports training-time summaries of optimization and region formation across sample sizes. Representative region and decision-boundary visualizations are shown in Figure 5.

**Exact region counts and training-time summaries.** For each seed and training checkpoint, we enumerate *exactly* the affine regions intersecting $\Omega$ using Algorithm 2. Thus, the entries in Table 2 are exact bounded-domain counts $|\mathcal{R}_\Omega(f)|$, aggregated over seeds. To complement these counts, Figure 4 summarizes four views of the same experiment across sample sizes: epoch vs. training accuracy, epoch vs. training loss, epoch vs. exact affine-region count, and exact affine-region count vs. training accuracy.

The exact counts in Table 2 indicate that the realized affine-region usage within $\Omega$ depends strongly on sample size under this protocol. For smaller sample sizes (e.g., 200 and 500), the exact region count tends to grow over training and reaches values well above initialization. For larger sample sizes (e.g., 5,000 and 10,000), the exact count drops sharply during training and remains far below the initialization level at later checkpoints. For example, at epoch 10,000, the reported count is $452 \pm 7$ for 5,000 samples and $68 \pm 6$ for 10,000 samples, compared with initialization levels near 1,151–1,152.

The training summaries in Figure 4 provide complementary context for these exact counts. In the reported runs, larger sample sizes are associated with different optimization trajectories in both training accuracy and training loss, while also exhibiting substantially smaller exact region counts. We therefore interpret the 2D random-label results conservatively: in this controlled setting, increasing sample size is associated with reduced realized affine-region usage within the bounded domain $\Omega$, with the relationship visible both in exact counts and in the corresponding training-time summaries.

**Ablation: initialization and optimizer.** To check that the observed collapse is not an artifact of a particular training configuration, we repeat the random-label experiment with 5000 samples and compare the exact region count at epoch 10000 under Xavier initialization and under SGD (keeping the remaining settings the same); the qualitative conclusion is unchanged, as all variants remain in the collapsed regime (see Table 3).

**Decision-boundary evolution versus boundary failure under complex inputs.** On a more regular, learnable dataset, the decision boundary evolves over training and progressively aligns with the data geometry; Figure 7 illustrates this behavior on `make_moons`, where both the exactly enumerated affine regions and the induced decision regions refine throughout training. In contrast, under random labels with 10000 samples, Figure 5 shows that the network fails to form an effective decision boundary, consistent with the concurrent reduction in realized affine regions reported in Table 2.

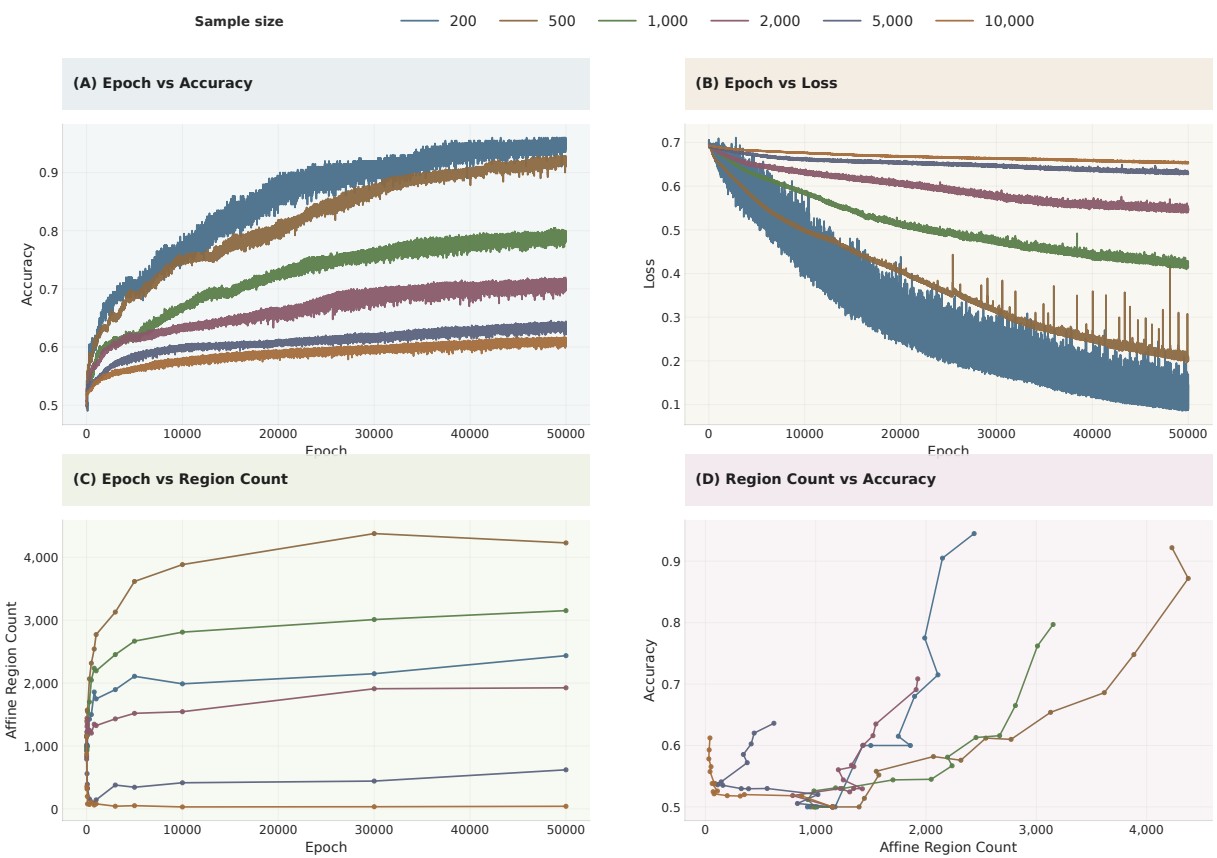

Figure 4: **Training summaries for the 2D random-label experiment with architecture** $[32, 32, 32]$, **aggregated over** $5$ **seeds.** Panels (A)–(D) report epoch vs. training accuracy, epoch vs. training loss, epoch vs. exact affine-region count in $\Omega = [-1, 1]^2$, and exact affine-region count vs. training accuracy, respectively, across different sample sizes. These plots complement Table 2 by summarizing how optimization and exact bounded-domain region counts evolve during training under increasing sample size.

### 6.4 High-dimensional exact enumeration of realized affine regions

We extend exact bounded-domain region enumeration to three-dimensional inputs and report *exact* counts of realized affine regions intersecting a compact domain. Because exact exploration in $d = 3$ is substantially more expensive than in the 2D setting, we focus here on a controlled random-label setup and summarize the configurations for which exact enumeration was completed.

**Dataset, domain, architectures, and reporting protocol.** We consider the random-data-with-random-labels setting in three dimensions ($d = 3$), where inputs are sampled uniformly from $\Omega = [-1, 1]^3$ and labels are assigned i.i.d. at random. We evaluate sample sizes 10,000, 30,000, 50,000, 70,000, and 100,000. We report results for fully-connected ReLU MLPs with hidden architectures $[16, 16, 16]$ and $[32, 32, 32]$, using the same training protocol as in the 2D experiments. Unless otherwise stated, all quantities in this subsection are aggregated over 5 random seeds and reported as mean $\pm$ standard deviation.

**Exact region enumeration across training.** At each reported epoch, we enumerate *exactly* the set of realized affine regions intersecting $\Omega$ using Algorithm 2. The entries in Table 4 therefore equal the exact bounded-domain counts $|\mathcal{R}_\Omega(f)|$ for the corresponding trained network and epoch.

**Statistical summaries of optimization and region formation.** To complement the exact region counts, Figure 6 summarizes training accuracy, training loss, affine-region count, and the relationship between

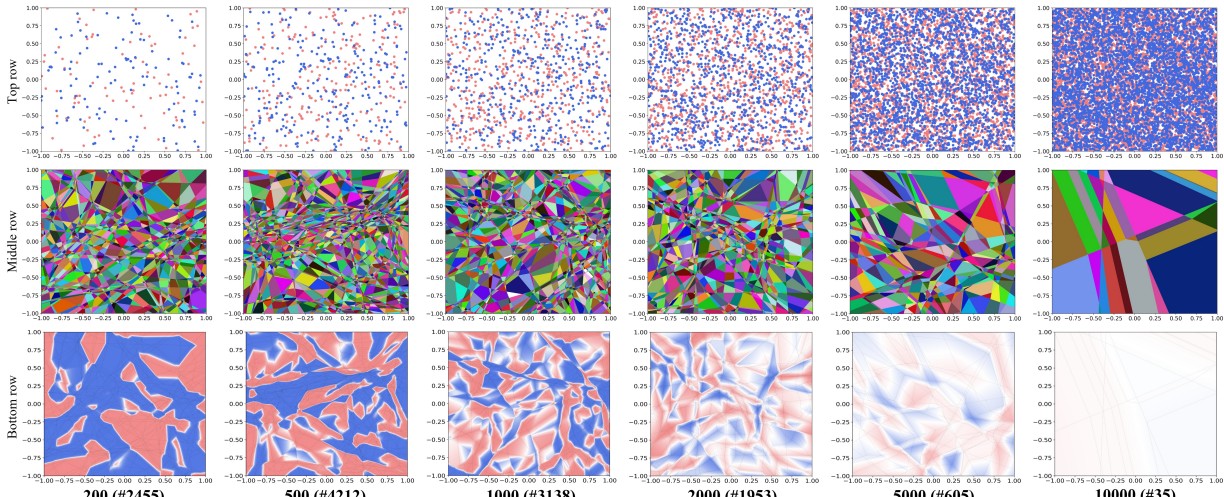

Figure 5: **Representative exact region and decision-boundary visualizations for random labels with increasing sample size (network** $[32, 32, 32]$**).** Top: training data. Middle: *exactly enumerated* affine regions in $\Omega$. Bottom: decision regions. These visualizations are intended to illustrate typical geometric patterns corresponding to the quantitative summaries in Table 2 and Figure 4.

Table 4: **Exact** realized affine-region counts $|\mathcal{R}_\Omega(f)|$ in $\Omega = [-1, 1]^3$ at selected training epochs for ReLU MLPs trained on random 3D inputs with random labels, computed by exhaustive bounded-domain exploration (Algorithm 2) and reported as mean $\pm$ standard deviation over 5 random seeds.

| Architecture | Samples | Epoch 0 | Epoch 10 | Epoch 30 | Epoch 50 | Epoch 80 | Epoch 100 |
|---|---|---|---|---|---|---|---|
| $[16, 16, 16]$ | 10,000 | $868_{\pm2}$ | $1,089_{\pm42}$ | $1,248_{\pm5}$ | $1,256_{\pm31}$ | $1,297_{\pm14}$ | $1,281_{\pm48}$ |
| $[16, 16, 16]$ | 30,000 | $867_{\pm3}$ | $885_{\pm27}$ | $970_{\pm44}$ | $1,062_{\pm12}$ | $1,067_{\pm35}$ | $1,160_{\pm6}$ |
| $[16, 16, 16]$ | 50,000 | $869_{\pm0}$ | $785_{\pm11}$ | $933_{\pm46}$ | $981_{\pm20}$ | $910_{\pm3}$ | $965_{\pm29}$ |
| $[16, 16, 16]$ | 70,000 | $868_{\pm2}$ | $723_{\pm41}$ | $856_{\pm8}$ | $825_{\pm33}$ | $726_{\pm15}$ | $695_{\pm47}$ |
| $[16, 16, 16]$ | 100,000 | $868_{\pm4}$ | $688_{\pm36}$ | $645_{\pm19}$ | $636_{\pm45}$ | $599_{\pm7}$ | $608_{\pm26}$ |
| $[32, 32, 32]$ | 10,000 | $12,656_{\pm3}$ | $10,084_{\pm13}$ | $9,906_{\pm37}$ | $10,342_{\pm4}$ | $11,639_{\pm28}$ | $12,209_{\pm16}$ |
| $[32, 32, 32]$ | 30,000 | $12,657_{\pm2}$ | $8,564_{\pm49}$ | $9,464_{\pm10}$ | $9,133_{\pm34}$ | $10,834_{\pm18}$ | $11,879_{\pm40}$ |
| $[32, 32, 32]$ | 50,000 | $12,658_{\pm0}$ | $9,609_{\pm30}$ | $8,520_{\pm47}$ | $8,629_{\pm22}$ | $8,722_{\pm1}$ | $8,579_{\pm38}$ |
| $[32, 32, 32]$ | 70,000 | $12,657_{\pm1}$ | $9,013_{\pm12}$ | $7,672_{\pm44}$ | $7,118_{\pm9}$ | $7,602_{\pm31}$ | $7,658_{\pm5}$ |
| $[32, 32, 32]$ | 100,000 | $12,657_{\pm0}$ | $8,082_{\pm14}$ | $6,953_{\pm27}$ | $6,251_{\pm3}$ | $6,631_{\pm41}$ | $6,163_{\pm20}$ |

affine-region count and training accuracy across epochs and sample sizes. In each block of the figure, the left and right subpanels correspond to architectures $[16, 16, 16]$ and $[32, 32, 32]$, respectively. Panels (A)–(D) report epoch vs. accuracy, epoch vs. loss, epoch vs. affine-region count, and affine-region count vs. accuracy.

The exact counts in Table 4 show that the 3D region dynamics depend on both architecture and sample size. For $[16, 16, 16]$, the final exact region counts at epoch 100 are above the initialization level for 10,000, 30,000, and 50,000 samples, but below initialization for 70,000 and 100,000 samples. For $[32, 32, 32]$, the final exact region counts decrease as the sample size increases over the reported range, from $12{,}209 \pm 16$ at 10,000 samples to $6{,}163 \pm 20$ at 100,000 samples.

# 7 Discussion

**Practical implications and scope.** These results suggest that realized affine-region usage may provide a useful perspective, beyond nominal worst-case capacity, when interpreting architecture and training behavior in controlled settings. In particular, our experiments show that, under a fixed architecture and training protocol, increasing task complexity need not translate into richer realized partitioning of the evaluation

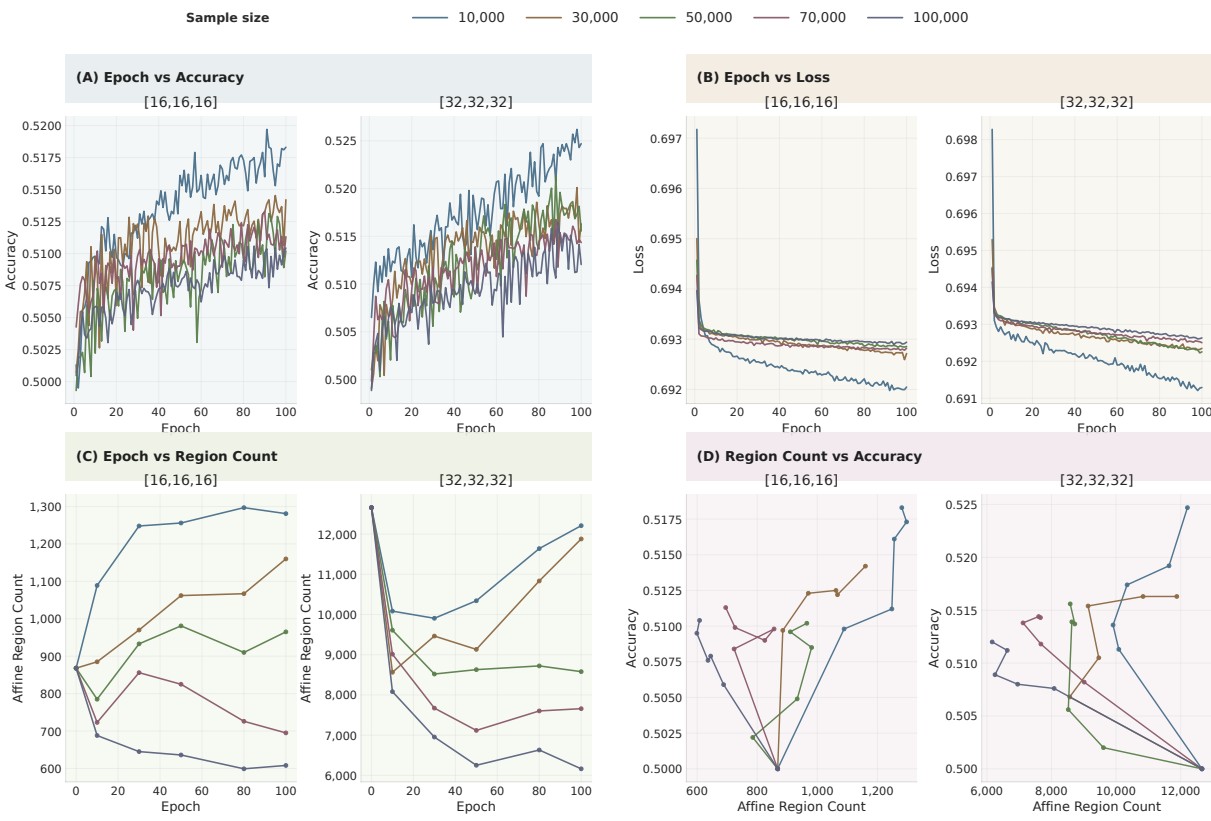

Figure 6: **Training summaries for the 3D random-label experiments.** Left and right subpanels within each block correspond to architectures $[16, 16, 16]$ and $[32, 32, 32]$, respectively. Panels (A)–(D) report epoch vs. training accuracy, epoch vs. training loss, epoch vs. exact affine-region count in $\Omega = [-1, 1]^3$, and exact affine-region count vs. training accuracy. These plots complement Table 4 by summarizing how optimization and exact bounded-domain region counts evolve across sample sizes and epochs.

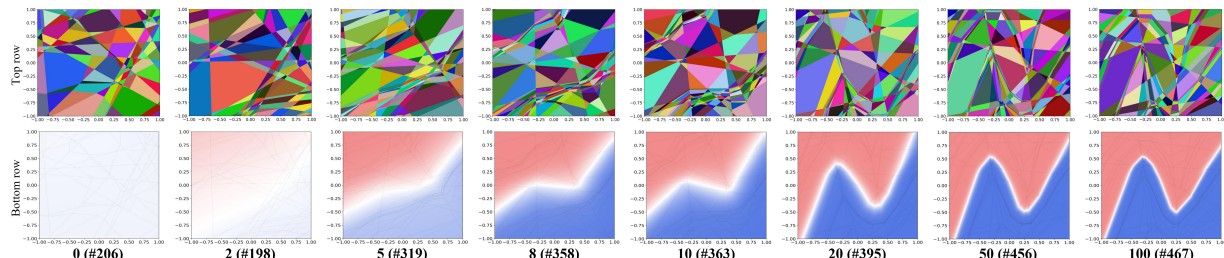

Figure 7: **Decision-boundary evolution on `make_moons`.** Network $[16, 16, 16]$. Top: *exactly enumerated* affine regions in $\Omega$. Bottom: decision regions. Epochs: $[0, 2, 5, 8, 10, 20, 50, 100]$.

domain. A practical implication is therefore interpretive rather than prescriptive: nominal expressive capacity alone may be an incomplete proxy for the nonlinear structure that is actually realized after training. From this perspective, region-usage statistics may be informative when comparing architectures or training settings in controlled regimes, especially when two models have similar nominal capacity but exhibit different realized geometric complexity after optimization.

At the same time, the present study is intentionally confined to fully-connected CPWA networks in low-dimensional settings where exact counting is feasible. Extending this perspective to more complex architectures,

models with non-piecewise-affine activations, and real-world datasets remains a primary direction for future work.

**Relation to existing empirical work on region utilization.** Our work is complementary to a broader literature, including empirical studies and theory-grounded analyses, on how neural networks utilize their piecewise-affine structure. Hanin & Rolnick (2019) study the complexity of affine regions in deep networks and show that practical affine-region complexity can remain far below worst-case bounds, including along one-dimensional subspaces. Zhang & Wu (2020) study local geometric properties of affine regions in trained networks and show that different optimization techniques can yield substantially different affine-region structures even when test accuracy is similar. Trimmel et al. (2021) examine how networks utilize affine regions for generalization by extracting data-dependent linear terms, and report substantial architectural differences in this utilization. Cohan et al. (2022) analyze region densities in deep reinforcement learning and find only moderate growth in observed complexity during training. Qi et al. (2023a) provide an empirical comparison of the realized numbers of affine regions in ReLU and LeakyReLU networks, highlighting how the choice of activation function can affect practically expressed piecewise-affine complexity. Qi et al. (2023b) study the effect of residual connections on the realized expressiveness of linear regions and report empirical differences in region utilization between residual and non-residual architectures. More recently, Patel & Montúfar (2025) formalize *local complexity* as a data-distribution-weighted density of affine regions and connect it to optimization and feature learning. At the same time, Gamba et al. (2022) argue that region density alone may fail to capture effective nonlinearity in modern overparameterized ReLU networks.

Compared with these studies, our paper examines realized affine-region usage under increasing task complexity for a fixed architecture and training protocol. Our empirical analysis is based on exact bounded-domain enumeration in 2D and higher dimensions, together with decision-boundary visualizations and training-time snapshots. This allows us to study how realized region usage and boundary geometry evolve in the controlled settings considered here. The empirical analysis is also paired with a rigorous 1D probe-level theorem and its neuron-threshold implication.

Our results are complementary to prior studies on region utilization. In the present setting, they highlight reduced realized affine-region usage under increasing task complexity and provide another view of the gap between expressive capacity and expressive utilization.

## 8 Limitations and future work

Our theoretical results are probe-based and do not directly characterize full-dimensional region counts or the geometry of regions away from one-dimensional trajectories. Exact bounded-domain enumeration, while exact, is computationally constrained and currently most practical in low dimensions and for moderate architectures. This limitation is not merely algorithmic. Exact affine-region enumeration for realistic complex structures and real-world datasets is closely related to a computationally intractable counting problem in general: recent complexity-theoretic results for ReLU networks show NP- and #P-hardness already in shallow settings, together with strong hardness-of-approximation results for deeper networks (Stargalla et al., 2025). Accordingly, the empirical conclusions of the present paper should be interpreted within the controlled low-dimensional settings where exact bounded-domain enumeration remains feasible.

Empirically, we focused on fully-connected ReLU MLPs and used random-label regimes as a controlled proxy for extreme task complexity. This design makes it possible to study the gap between expressive capacity and realized affine-region usage under fixed architectures and training protocols, but it does not by itself establish how the same phenomena manifest in more structured data or in real-world applications. Extending the analysis to broader architectures, including convolutional, residual, and attention-based models, as well as to more structured notions of data complexity, remains an important direction for future work.

More broadly, several directions remain open. These include developing theory for region formation under specific training dynamics, clarifying how realized affine-region usage relates to known optimization phenomena, designing regularization or curriculum strategies that stabilize region refinement under complex supervision, and developing exact, approximate, or certified methods that can scale to data manifolds and higher-dimensional domains.

# 9 Conclusion

We study the gap between affine-region capacity and the regions realized after training in piecewise-affine networks. We prove an architecture-dependent upper bound on affine pieces along any affine line-segment probe, yielding a necessary neuron threshold for 1D piece complexity. In the controlled low-dimensional fully-connected ReLU MLP settings studied here, exact 1D counting and bounded-domain enumeration in 2D and 3D provide evidence of *expressivity saturation*: increasing task complexity in our random-label/sample-size regimes is associated with fewer realized regions and, in 2D, degraded decision boundaries.

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

# A Proofs

## A.1 Proof of Theorem 5

Throughout this subsection, we consider a *line-segment probe*

$$\gamma(t) \; = \; x_0 + t\,v, \qquad t \in [0, 1], \tag{14}$$

for some $x_0, v \in \mathbb{R}^d$. This is the standard setting for line-restricted region counting.

**CPWA activation.** Let $\sigma : \mathbb{R} \to \mathbb{R}$ be CPWA with (finite) breakpoint set

$$-\infty = t_0 < t_1 < \cdots < t_K < t_{K+1} = +\infty, \tag{15}$$

such that for each $k \in \{1, \ldots, K+1\}$ there exist scalars $c_k, d_k \in \mathbb{R}$ with

$$\sigma(z) \; = \; c_k z + d_k, \qquad z \in (t_{k-1}, t_k). \tag{16}$$

**Network notation.** Define the hidden layers recursively by

$$h_0(x) := x, \tag{17}$$
$$h_\ell(x) := \sigma\big(W_\ell h_{\ell-1}(x) + b_\ell\big), \qquad \ell = 1, \ldots, L, \tag{18}$$
$$f(x) := W_{L+1} h_L(x) + b_{L+1}, \tag{19}$$

where $\sigma$ acts coordinate-wise.

**Piece count along the probe.** For each $\ell \in \{0, 1, \ldots, L\}$, let $R_\ell$ be the smallest integer such that there exists a partition

$$0 = \tau_0 < \tau_1 < \cdots < \tau_{R_\ell} = 1 \tag{20}$$

for which $t \mapsto h_\ell(\gamma(t))$ is affine on every open interval $(\tau_{r-1}, \tau_r)$. Since the output layer is affine in $h_L$, any partition witnessing affine behavior for $h_L \circ \gamma$ also witnesses affine behavior for $f \circ \gamma$, hence

$$\alpha(f; \gamma) \; \leq \; R_L. \tag{21}$$

We now prove a layer-wise refinement bound.

**Lemma 9** (Affine pre-activations on each incoming affine piece). *Fix $\ell \in \{1, \ldots, L\}$ and an open interval $I \subset (0, 1)$. If $t \mapsto h_{\ell-1}(\gamma(t))$ is affine on $I$, then for each neuron $j \in \{1, \ldots, n_\ell\}$ the pre-activation*

$$a_{\ell,j}(t) \; := \; e_j^\top\big(W_\ell h_{\ell-1}(\gamma(t)) + b_\ell\big) \tag{22}$$

*is an affine function of $t$ on $I$.*

*Proof.* If $h_{\ell-1}(\gamma(t))$ is affine on $I$, then there exist $u, v \in \mathbb{R}^{n_{\ell-1}}$ such that

$$h_{\ell-1}(\gamma(t)) \; = \; u + t\,v, \qquad t \in I. \tag{23}$$

Substituting equation 23 into equation 22 yields

$$a_{\ell,j}(t) \; = \; e_j^\top(W_\ell u + b_\ell) \; + \; t\,e_j^\top(W_\ell v), \tag{24}$$

which is affine in $t$ on $I$. $\qquad \square$

**Lemma 10** (A CPWA scalar neuron introduces at most $K$ split points on an affine input). *Let $I \subset \mathbb{R}$ be a nonempty open interval and let $a : I \to \mathbb{R}$ be affine. Then $\sigma \circ a$ is piecewise-affine on $I$ and there exists a partition of $I$ into at most $K+1$ open subintervals on each of which $\sigma \circ a$ is affine. Equivalently, the affine formula of $\sigma \circ a$ changes at most $K$ times on $I$.*

*Proof.* Write $a(t) = \alpha t + \beta$ on $I$. For each breakpoint $t_k$ in equation 15, consider the equation

$$a(t) = t_k. \tag{25}$$

If $\alpha \neq 0$, then equation 25 has at most one solution in $I$ for each $k \in \{1, \ldots, K\}$. If $\alpha = 0$, then $a$ is constant on $I$ and $\sigma \circ a$ is constant (hence affine) on $I$. Define the set of potential split points

$$\mathcal{Z} := \{t \in I : a(t) \in \{t_1, \ldots, t_K\}\}. \tag{26}$$

From the above discussion,

$$|\mathcal{Z}| \leq K. \tag{27}$$

On any connected component of $I \setminus \mathcal{Z}$, the value $a(t)$ lies within a single interval $(t_{k-1}, t_k)$, hence $\sigma$ follows a fixed affine rule equation 16 and $\sigma \circ a$ is affine there. Removing at most $K$ points from an open interval produces at most $K + 1$ open components, proving the claim. □

**Lemma 11** (One layer of width $n_\ell$ refines each incoming affine piece by at most $Kn_\ell$ splits). *Fix $\ell \in \{1, \ldots, L\}$ and an open interval $I \subset (0, 1)$. If $t \mapsto h_{\ell-1}(\gamma(t))$ is affine on $I$, then there exists a partition of $I$ into at most $Kn_\ell + 1$ open subintervals on each of which $t \mapsto h_\ell(\gamma(t))$ is affine.*

*Proof.* By Lemma 9, each pre-activation $a_{\ell,j}$ is affine on $I$. By Lemma 10, for each $j$ there is a set $\mathcal{Z}_j \subset I$ with

$$|\mathcal{Z}_j| \leq K \tag{28}$$

such that $\sigma \circ a_{\ell,j}$ is affine on every connected component of $I \setminus \mathcal{Z}_j$. Let

$$\mathcal{Z} := \bigcup_{j=1}^{n_\ell} \mathcal{Z}_j. \tag{29}$$

Then

$$|\mathcal{Z}| \leq \sum_{j=1}^{n_\ell} |\mathcal{Z}_j| \leq Kn_\ell. \tag{30}$$

On any connected component of $I \setminus \mathcal{Z}$, every coordinate $\sigma(a_{\ell,j}(t))$ follows a fixed affine branch, so there exist a diagonal matrix $C \in \mathbb{R}^{n_\ell \times n_\ell}$ and a vector $d \in \mathbb{R}^{n_\ell}$ (constant on that component) with

$$\sigma\big(W_\ell h_{\ell-1}(\gamma(t)) + b_\ell\big) = C\big(W_\ell h_{\ell-1}(\gamma(t)) + b_\ell\big) + d. \tag{31}$$

Since $h_{\ell-1}(\gamma(t))$ is affine on $I$, the right-hand side of equation 31 is affine on that component. Finally, removing at most $Kn_\ell$ points from an open interval yields at most $Kn_\ell + 1$ open components, completing the proof. □

*Proof of Theorem 5.* We prove by induction that

$$R_\ell \leq \prod_{s=1}^{\ell} (Kn_s + 1), \qquad \ell \in \{0, 1, \ldots, L\}, \tag{32}$$

with the convention that the empty product equals 1.

**Base case.** By equation 14 and equation 17, the map $t \mapsto h_0(\gamma(t))$ is affine on $(0, 1)$, hence

$$R_0 = 1, \tag{33}$$

which matches equation 32 for $\ell = 0$.

**Inductive step.** Fix $\ell \in \{1, \ldots, L\}$ and assume equation 32 holds for $\ell - 1$. Let $[0, 1]$ be partitioned into $R_{\ell-1}$ intervals as in equation 20 such that $h_{\ell-1} \circ \gamma$ is affine on each open interval. On each such open interval, Lemma 11 gives a refinement into at most $Kn_\ell + 1$ open subintervals on which $h_\ell \circ \gamma$ is affine. Therefore

$$R_\ell \ \leq \ (Kn_\ell + 1)\, R_{\ell-1}. \tag{34}$$

Combining equation 34 with the inductive hypothesis yields equation 32 for $\ell$.

**Conclusion.** Taking $\ell = L$ in equation 32 gives

$$R_L \ \leq \ \prod_{\ell=1}^{L} (Kn_\ell + 1). \tag{35}$$

Using equation 21, we obtain

$$\alpha(f; \gamma) \ \leq \ \prod_{\ell=1}^{L} (Kn_\ell + 1), \tag{36}$$

which is the desired bound. $\qquad\square$

### A.2 Proof of Corollary 7

*Proof.* Assume $f(\gamma(s)) = g(s)$ for all $s \in [0, 1]$. By definition of $Q(g)$, any partition of $[0, 1]$ into intervals on which $g$ is affine must have at least $Q(g)$ intervals. Since $g = f \circ \gamma$, any partition witnessing $\alpha(f; \gamma)$ also witnesses that $g$ is affine on each piece. Hence

$$Q(g) \ \leq \ \alpha(f; \gamma). \tag{37}$$

Applying Theorem 5 yields

$$\alpha(f; \gamma) \ \leq \ \prod_{\ell=1}^{L} (Kn_\ell + 1), \tag{38}$$

and therefore

$$\prod_{\ell=1}^{L} (Kn_\ell + 1) \ \geq \ Q(g). \tag{39}$$

Let $N := \sum_{\ell=1}^{L} n_\ell$. By AM–GM applied to the positive numbers $(Kn_\ell + 1)_{\ell=1}^{L}$,

$$\prod_{\ell=1}^{L} (Kn_\ell + 1) \ \leq \ \left( \frac{1}{L} \sum_{\ell=1}^{L} (Kn_\ell + 1) \right)^{L}. \tag{40}$$

Compute the mean:

$$\frac{1}{L} \sum_{\ell=1}^{L} (Kn_\ell + 1) = \frac{1}{L} \left( K \sum_{\ell=1}^{L} n_\ell + \sum_{\ell=1}^{L} 1 \right), \tag{41}$$

$$= \frac{1}{L} (KN + L), \tag{42}$$

$$= 1 + \frac{KN}{L}. \tag{43}$$

Combining equation 39, equation 40, and equation 43 gives

$$Q(g) \ \leq \ \left( 1 + \frac{KN}{L} \right)^{L}. \tag{44}$$

Taking $L$-th roots and rearranging yields

$$N \ \geq \ \frac{L}{K}\big(Q(g)^{1/L} - 1\big). \tag{45}$$

If $n_\ell = w$ for all $\ell$, then equation 39 becomes

$$(Kw + 1)^L \ \geq \ Q(g), \tag{46}$$

equivalently

$$w \ \geq \ \frac{Q(g)^{1/L} - 1}{K}. \tag{47}$$

Since $w$ is an integer, we obtain

$$w \ \geq \ \left\lceil \frac{Q(g)^{1/L} - 1}{K} \right\rceil, \tag{48}$$

as claimed. $\qquad\square$

# B    Residual-Connection Ablation under Complex 2D Inputs

This appendix reports an additional ablation for the 2D random-label setting in which the baseline MLP is augmented with a single residual connection. Under the same data and training protocol as in the main 2D experiments, we compare exact bounded-domain affine-region counts and summarize the corresponding optimization trajectories.

## B.1    Setup

**Data and task.**    We consider 2D random inputs with random labels, with inputs normalized to the same bounded domain as in the main text, $\Omega = [-1, 1]^2$. We sweep the number of training samples over

$$N_{\text{data}} \in \{200, 500, 1000, 2000, 3000, 5000, 7000, 10000\},$$

and train using cross-entropy loss for classification, matching the main experimental protocol.

**Optimization and reporting protocol.**    All runs use Adam, batch size 32, and are trained for 10000 epochs. Unless otherwise stated, all remaining hyperparameters and initialization follow Section 6.1 exactly. To quantify variability, all scalar summaries in this appendix are aggregated over 5 random seeds and reported as mean $\pm$ standard deviation.

**Architectures with a single residual connection.**    We consider three depth-$L = 3$ ReLU MLP backbones with widths $[16, 16, 16]$, $[32, 32, 32]$, and $[64, 64, 64]$, and augment each with one residual connection. Concretely, denoting the hidden representations by

$$h_1(x) = \sigma(W_1 x + b_1), \tag{49}$$
$$h_2(x) = \sigma(W_2 h_1(x) + b_2), \tag{50}$$
$$\tilde{h}_3(x) = \sigma(W_3 h_2(x) + b_3), \tag{51}$$

we define the residualized third hidden layer as

$$h_3(x) \; = \; \tilde{h}_3(x) + h_1(x), \tag{52}$$

i.e., a single identity skip from the first to the third hidden layer. The output layer remains linear:

$$f(x) = W_4 h_3(x) + b_4. \tag{53}$$

This construction preserves dimensional compatibility because all hidden layers have equal width within each backbone.

**Exact region counting and training summaries.**    As in the main text, the reported region numbers are *exact* counts of realized affine regions intersecting $\Omega$, computed via exhaustive bounded-domain region exploration (Algorithm 2). For each configuration, Table 5 reports the exact region count in $\Omega$ after training to epoch 10000. In addition, Figure 8 summarizes the corresponding optimization trajectories by plotting epoch vs. training accuracy and epoch vs. training loss across sample sizes.

## B.2    Results

**Exact realized region counts.**    Table 5 reports the exact number of affine regions intersecting $\Omega = [-1, 1]^2$ for each residualized architecture and sample size at epoch 10000, aggregated over 5 seeds. In the reported runs, the final exact region count depends on both architecture width and sample size. For all three residualized architectures, the counts are comparatively larger at smaller sample sizes and substantially smaller at the largest sample sizes considered here. At 10000 samples, for example, the exact counts are $48 \pm 7$, $115 \pm 12$, and $98 \pm 15$ for $[16, 16, 16]$, $[32, 32, 32]$, and $[64, 64, 64]$ with a residual connection, respectively.

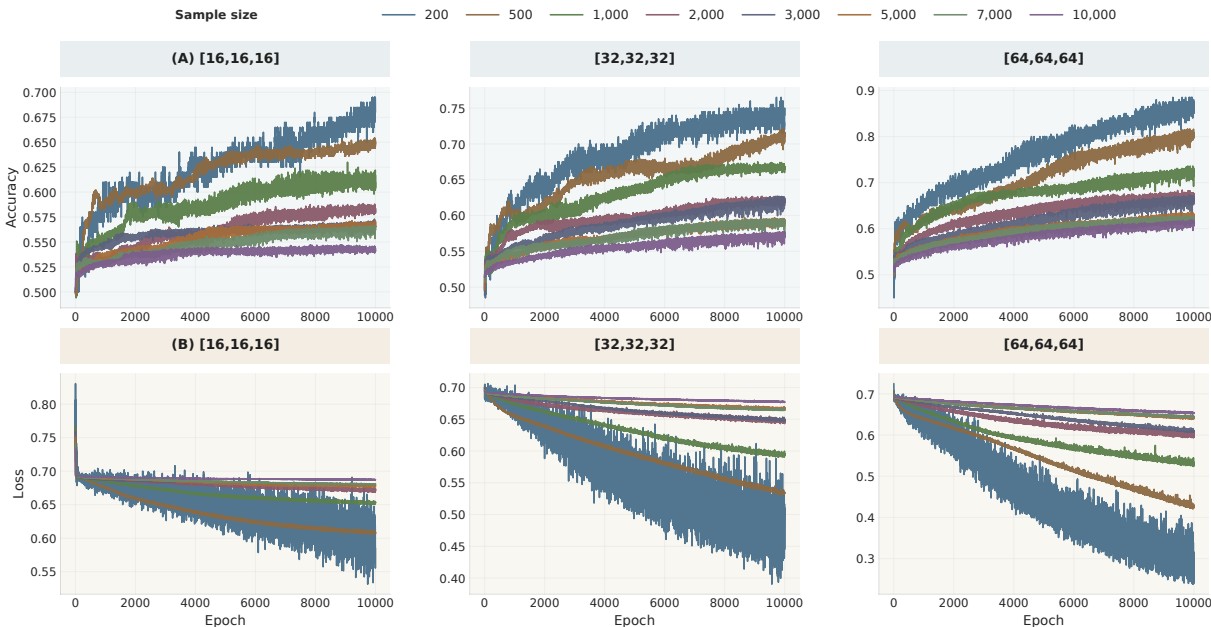

Figure 8: **Training summaries for the residualized 2D random-label experiments, aggregated over** 5 **seeds.** The figure reports epoch vs. training accuracy and epoch vs. training loss across sample sizes for the residualized architectures considered in this appendix. These summaries complement Table 5 by showing the corresponding optimization trajectories for the same experimental setting.

Table 5: **Exact** realized affine-region counts $|\mathcal{R}_\Omega(f)|$ in $\Omega = [-1, 1]^2$ at epoch 10000 for residualized ReLU MLPs trained on random 2D inputs with random labels, reported as mean $\pm$ standard deviation over 5 random seeds. All settings match the main experiments; only the architecture differs by adding one residual connection as in equation 52.

| Architecture (with one residual) | 200 | 500 | 1000 | 2000 | 3000 | 5000 | 7000 | 10000 |
|---|---|---|---|---|---|---|---|---|
| $[16, 16, 16]$ + residual | $2049_{\pm 17}$ | $2176_{\pm 43}$ | $2189_{\pm 8}$ | $1874_{\pm 31}$ | $853_{\pm 24}$ | $512_{\pm 46}$ | $347_{\pm 13}$ | $48_{\pm 7}$ |
| $[32, 32, 32]$ + residual | $3748_{\pm 29}$ | $5597_{\pm 6}$ | $3793_{\pm 41}$ | $2147_{\pm 18}$ | $2908_{\pm 35}$ | $741_{\pm 9}$ | $911_{\pm 48}$ | $115_{\pm 12}$ |
| $[64, 64, 64]$ + residual | $7613_{\pm 12}$ | $9050_{\pm 44}$ | $7421_{\pm 27}$ | $6879_{\pm 5}$ | $2683_{\pm 39}$ | $1287_{\pm 16}$ | $594_{\pm 33}$ | $98_{\pm 15}$ |

**Training accuracy and loss trajectories.** Figure 8 complements the exact region counts by reporting training accuracy and training loss over epochs for the same residualized models across different sample sizes. These plots provide optimization context for the final exact region counts in Table 5 and show that the optimization trajectories vary across both sample size and architecture. In this controlled random-label setting and under the reported protocol, adding a single residual connection therefore does not by itself remove the tendency toward smaller final realized affine-region counts at larger sample sizes within the bounded domain $\Omega$.

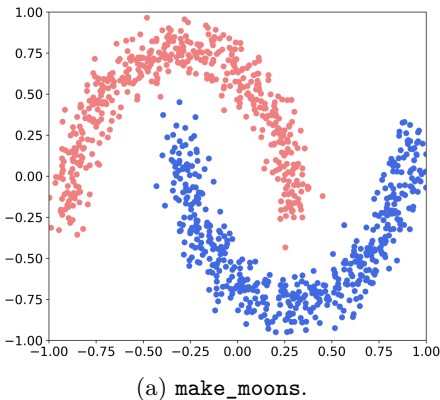
(a) make_moons.

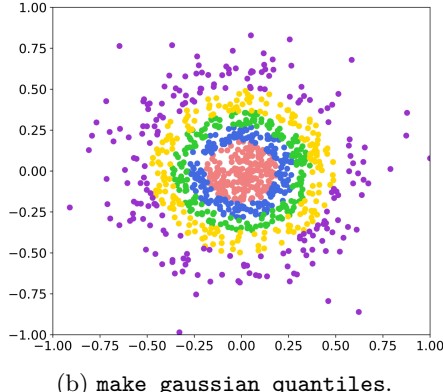
(b) make_gaussian_quantiles.

Figure 9: Two synthetic datasets used for visualizing training dynamics. **Left:** the make_moons dataset with 1000 samples and 2 classes. **Right:** the make_gaussian_quantiles dataset with 1000 samples and 5 classes.

## C    Additional Visualizations of Training Dynamics

This appendix provides additional visualizations of the training dynamics discussed in the main paper. The goal is to complement the quantitative region-count results with qualitative and temporal evidence of how affine partitions and decision boundaries evolve during optimization. Unless otherwise stated, the experimental protocol follows the setup described in Section 6.1: we use fully-connected ReLU MLPs, train with the same optimizer and loss function, and count affine regions within the bounded domain $\Omega = [-1, 1]^2$. The results in this appendix are intended as supplementary visual evidence rather than as separate claims beyond the main experiments.

### C.1    Datasets

We consider two standard two-dimensional synthetic datasets for visualizing the evolution of affine-region partitions and decision boundaries. The first dataset, make_moons, provides a binary classification problem with a nonlinearly separable but geometrically structured decision boundary. The second dataset, make_gaussian_quantiles, contains five classes arranged according to Gaussian quantiles, leading to a more multi-class decision structure. These datasets are useful for visual inspection because the input space is two-dimensional, allowing both affine regions and decision regions to be plotted directly.

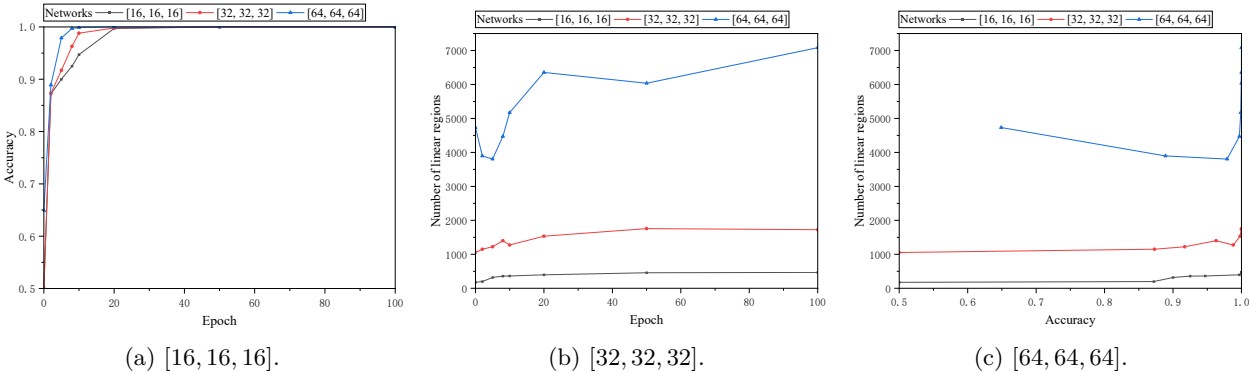

(a) [16, 16, 16].  (b) [32, 32, 32].  (c) [64, 64, 64].

Figure 10: Training dynamics on the `make_moons` dataset in Figure 9a. Each panel shows the relationship among affine-region count, accuracy, and training epoch for a different network architecture.

Table 6: Affine-region counts across training epochs on the `make_moons` dataset in Figure 9a.

| Epoch
DNNs | 0 | 2 | 5 | 8 | 10 | 20 | 50 | 100 |
|---|---|---|---|---|---|---|---|---|
| [16, 16, 16] | 206 | 198 | 319 | 358 | 363 | 395 | 456 | 467 |
| [32, 32, 32] | 1151 | 1151 | 1222 | 1401 | 1274 | 1533 | 1756 | 1725 |
| [64, 64, 64] | 4112 | 3897 | 3803 | 4472 | 5172 | 6354 | 6037 | 7082 |

## C.2 Region-count and accuracy curves during training

We first examine the temporal relationship between affine-region counts and classification accuracy. For each architecture, we track the number of realized affine regions and the training accuracy across selected epochs. These curves provide a complementary view of the region-refinement process: during training, the network may increase the number of realized affine regions as it adapts its piecewise-affine partition to the data geometry. However, the relationship between region count and accuracy is not strictly monotone, since region formation also depends on optimization dynamics and on how the regions are positioned relative to the class boundaries.

For the `make_moons` dataset, the affine-region counts generally increase during training for all three architectures, as summarized in Table 6. This trend is consistent with the visual intuition that the networks progressively refine their piecewise-affine partitions in order to represent the curved binary decision boundary. The larger architectures start with and maintain substantially more regions than the smaller architecture, reflecting their larger architectural capacity. At the same time, the counts are not perfectly monotone across epochs.

For the `make_gaussian_quantiles` dataset, the region counts also increase substantially during training, especially for the wider networks. Compared with the binary `make_moons` task, this dataset involves a multiclass decision structure, and the learned partitions become more refined as training proceeds. The increase is particularly pronounced for the [64, 64, 64] network, whose region count grows from 4113 at initialization to 8579 at epoch 100. These observations support the interpretation that, on learnable structured datasets, training can actively allocate additional affine regions to fit the geometry of the target decision function.

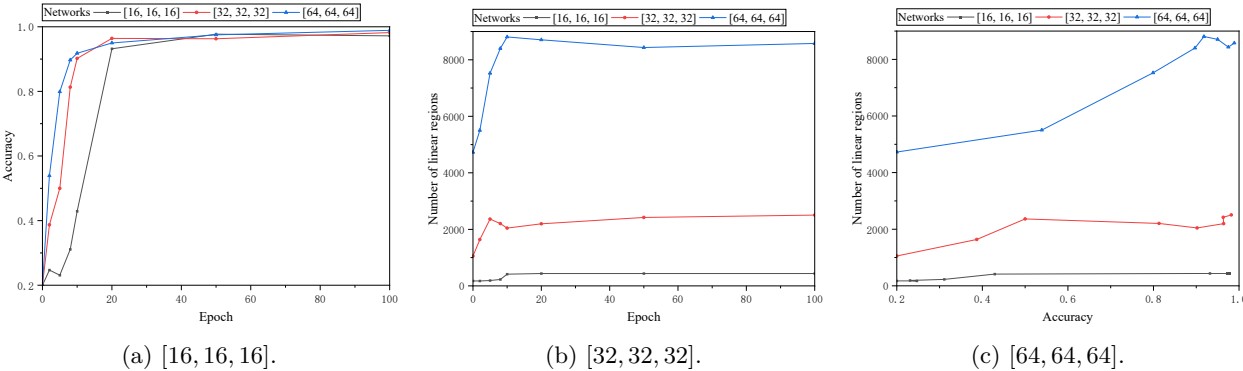

(a) $[16, 16, 16]$.      (b) $[32, 32, 32]$.      (c) $[64, 64, 64]$.

Figure 11: Training dynamics on the `make_gaussian_quantiles` dataset in Figure 9b. Each panel shows the relationship among affine-region count, accuracy, and training epoch for a different network architecture.

Table 7: Affine-region counts across training epochs on the `make_gaussian_quantiles` dataset in Figure 9b.

| Epoch
DNNs | 0 | 2 | 5 | 8 | 10 | 20 | 50 | 100 |
|---|---|---|---|---|---|---|---|---|
| $[16, 16, 16]$ | 207 | 174 | 189 | 228 | 415 | 437 | 438 | 436 |
| $[32, 32, 32]$ | 1154 | 1638 | 2362 | 2204 | 2045 | 2197 | 2420 | 2505 |
| $[64, 64, 64]$ | 4113 | 5501 | 7524 | 8402 | 8809 | 8706 | 8433 | 8579 |

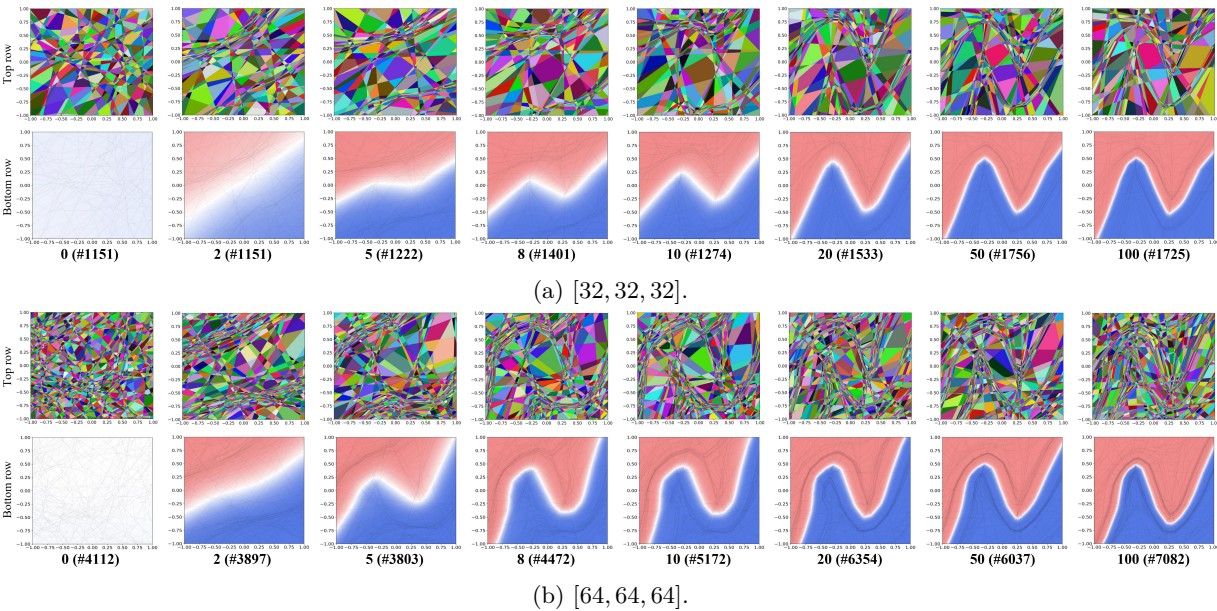

(a) $[32, 32, 32]$.

(b) $[64, 64, 64]$.

Figure 12: Evolution of affine-region partitions and decision regions on the `make_moons` dataset in Figure 9a. The selected epochs are $0, 2, 5, 8, 10, 20, 50$, and $100$. For each architecture, the top row shows the affine-region partition and the bottom row shows the decision regions. Different colors indicate predicted classes, and the transition bands between colors indicate the learned decision boundaries.

## C.3 Evolution of decision boundaries

We further visualize how the affine partitions and decision boundaries evolve over training. For each selected epoch, the upper visualization shows the affine-region partition induced by the trained network within the evaluation domain, while the lower visualization shows the corresponding decision regions. These plots make it possible to inspect not only how many affine regions are present, but also how they are spatially organized relative to the data distribution. In particular, they illustrate that successful training is not merely associated with having many affine regions; rather, the regions must be arranged in a way that supports an appropriate decision boundary.

On the `make_moons` dataset, the decision-boundary visualizations show a gradual refinement process. At early epochs, the decision regions are relatively coarse and do not yet align well with the curved data geometry. As training progresses, the affine partition becomes more structured, and the decision boundary increasingly follows the two-moon shape. This provides qualitative evidence that the increase in affine-region usage reported in Table 6 is accompanied by a meaningful reorganization of the decision geometry.

On the `make_gaussian_quantiles` dataset, the learned decision regions become increasingly structured over training and reflect the multi-class organization of the data. The wider networks produce finer affine partitions and more detailed class boundaries, consistent with the larger region counts in Table 7. Together with the `make_moons` results, these visualizations support the view that, in structured and learnable settings, optimization can refine both the affine-region partition and the corresponding decision boundary over time.

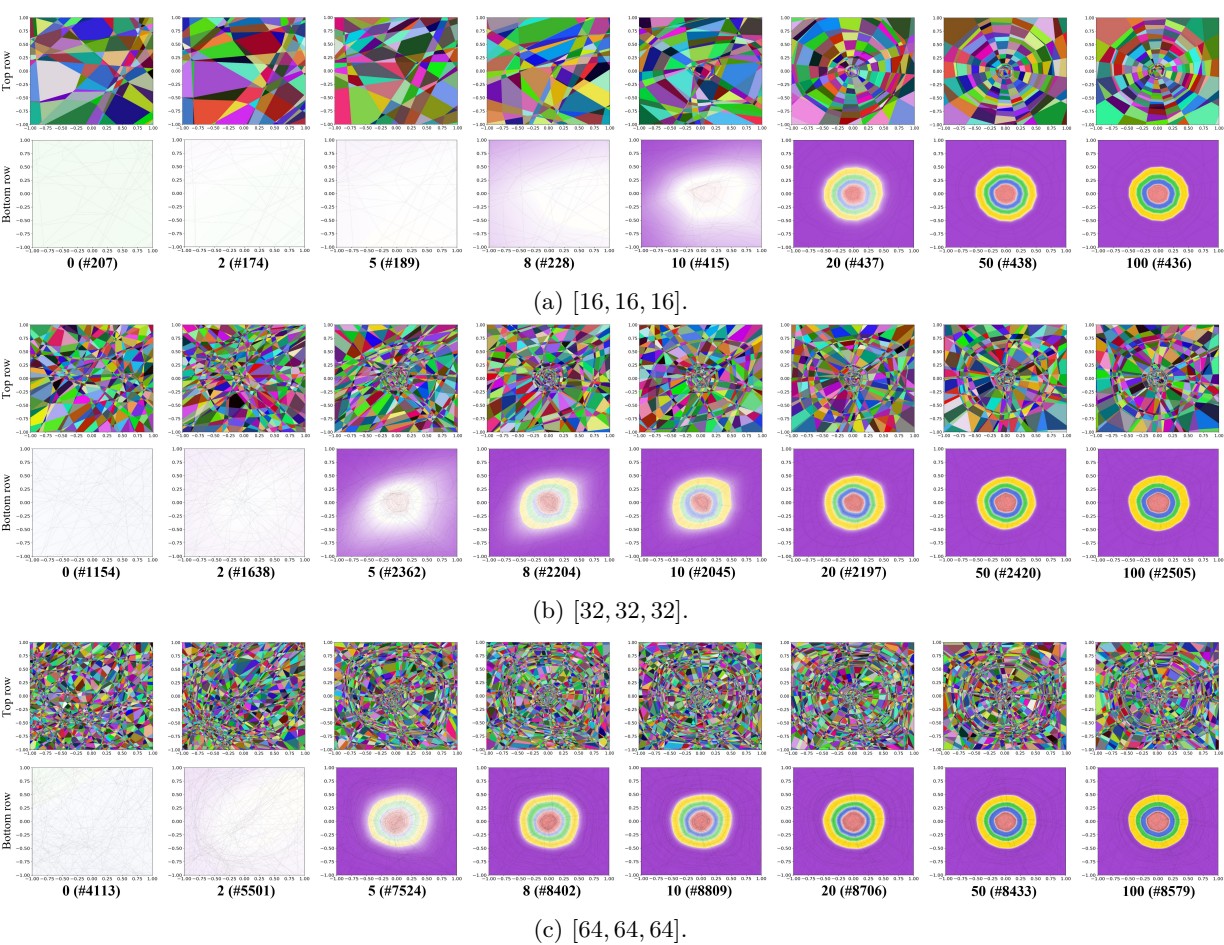

(a) $[16, 16, 16]$.

(b) $[32, 32, 32]$.

(c) $[64, 64, 64]$.

Figure 13: Evolution of affine-region partitions and decision regions on the `make_gaussian_quantiles` dataset in Figure 9b. The selected epochs are $0, 2, 5, 8, 10, 20, 50,$ and $100$. For each architecture, the top row shows the affine-region partition and the bottom row shows the decision regions. Different colors indicate predicted classes, and the transition bands between colors indicate the learned decision boundaries.

