# OpenReview forum: "Expressivity Saturation: Reduced Affine Region Usage Under Increasing Task Complexity"
_TMLR — Accepted by TMLR_

### Review · Reviewer_x9Rf · 2026-03-27

**Summary Of Contributions:**

The paper examines the gap between the theoretical expressive capacity of piecewise-affine neural networks and the number of affine regions they actually employ after training. It introduces a rigorous theorem that establishes an architecture-dependent upper bound for one-dimensional probes, determining the minimum number of neurons needed to model complex signals. By performing exact enumerations in two- and three-dimensional settings, the authors uncover a phenomenon termed expressivity saturation, in which increasing task complexity results in a pronounced decline in the number of realized affine regions.

**Audience:**

Yes

**Audience Explanation:**

The deep learning and ML audience will benefit from this research as it addresses fundamental questions regarding the expressive capacity and practical utilization of neural networks through a combination of rigorous theory and novel empirical observations

**Broader Impact Concerns:**

There is no concern for ethical implications.

**Claims And Evidence:**

Yes

**Claims Explanation:**

The evidence presented in the sources is accurate, clearly articulated, and supported by both rigorous mathematical analysis and extensive empirical results. The work explicitly defines its scope and limitations, acknowledging that while the findings are reliable for the tested MLP architectures and controlled settings, further research is required to extend these exact counting methods to more complex models such as Transformers. By including the complete procedures (Algorithms 1 and 2) used for counting, the authors promote transparency and reproducibility.

**Requested Changes:**

At this stage, no changes appear essential for supporting a recommendation of acceptance, as the authors have already substantiated their main claim with rigorous proofs, precise counting methods, and comprehensive ablation studies, including analyses of initialization, optimization strategies, and residual connections.

---

> ### Author Response · Authors · 2026-03-27
> **Response to Reviewer x9Rf**
>
> We thank the reviewer for the careful reading and positive assessment of our submission.
>
> We appreciate the recognition of the paper’s theoretical contributions, exact counting methodology, and empirical validation, as well as the acknowledgement that the scope and limitations are clearly stated. We are also grateful for the reviewer’s comment that no essential changes appear necessary at this stage.
>
> We will continue to improve the clarity and presentation of the paper in the revision. Thank you again for your time and constructive feedback.

---

### Review · Reviewer_NaUw · 2026-04-06

**Summary Of Contributions:**

Summary of Contributions:

The paper studies the gap between the theoretical expressivity of piecewise-affine neural networks (measured via affine region counts) and the expressivity actually realized after training. It provides both theoretical and empirical contributions.
On the theoretical side, the paper derives an architecture dependent upper bound on the number of affine regions along one-dimensional probes, leading to a corresponding neuron-threshold lower bound for representing functions with given piecewise complexity.
On the empirical side, the paper proposes exact enumeration methods for affine regions in bounded domains (1D, 2D, and higher dimensions), and demonstrates a phenomenon termed expressivity saturation, where increasing task complexity leads to a decrease in the number of realized regions under fixed architectures and training protocols. The work further links this region collapse to degraded decision boundaries and studies the dynamics of region formation during training.

Strengths:
1. Clear and well-motivated problem (gap between capacity and utilization).

2. Non-trivial theoretical result with an explicit, interpretable bound.

3. Exact (not approximate) region counting methodology.

4. Interesting empirical observation (expressivity saturation) with visual and quantitative support.

Weaknesses:

1. Theoretical results are limited to 1D probes.

2. Empirical settings rely heavily on synthetic/random-label regimes.

3. Practical implications for modern architectures (e.g., CNNs, Transformers) remain unclear.

**Audience:**

Yes

**Audience Explanation:**

The paper addresses an important question about neural network expressivity and optimization: the discrepancy between capacity and utilization. This is of interest to the TMLR audience, especially those working on theory, geometry of neural networks, and training dynamics. The combination of theory and exact empirical analysis makes it particularly relevant.

**Broader Impact Concerns:**

The work does not raise immediate societal risks.

**Claims And Evidence:**

Yes

**Claims Explanation:**

The theoretical claims are rigorously derived and internally consistent, with proofs provided in the appendix. The empirical claims are supported by exact region counting, which strengthens credibility. The experiments systematically vary task complexity and show consistent trends across settings (e.g., region collapse with increasing sample size in 2D and 3D).
However, the evidence is primarily limited to controlled synthetic settings (e.g., random labels), which raises questions about how broadly the conclusions transfer to realistic datasets and tasks.

**Requested Changes:**

1. Please discuss how findings (especially expressivity saturation) are expected to transfer beyond synthetic/random-label settings to real-world datasets.

2.  It should be clearer whether “expressivity saturation” is a fundamentally new phenomenon or a reinterpretation of known optimization limitations (e.g., underfitting, implicit bias).

3. The gap between 1D theoretical guarantees and higher-dimensional empirical observations should be better articulated.

4. Please add comparisons or discussion relative to existing empirical works on region utilization.

5. Please improve clarity around practical implications (e.g., how one might use these findings in model design or training).

---

> ### Author Response · Authors · 2026-04-19
> **Response to Reviewer NaUw**
>
> We thank the reviewer NaUw for the careful reading and constructive suggestions. We have revised the manuscript accordingly and clarify the requested points below.
>
> 1. On transfer beyond synthetic/random-label settings.
>
> We have clarified this point in the revised manuscript by adding explicit discussion in the Limitations section, marked in magenta. In particular, we now state more clearly that exact affine-region enumeration is extremely difficult for realistic complex structures and real-world datasets, and is computationally intractable in general: recent complexity-theoretic results show NP- and #P-hardness even in shallow ReLU settings. Accordingly, we make the scope of our conclusions and the limits of transfer beyond the controlled low-dimensional settings studied here more explicit.
>
> 2. On whether “expressivity saturation” is a fundamentally new phenomenon or a reinterpretation of known optimization limitations.
>
> We have clarified this point in the revised manuscript by adding an explicit sentence in Section 1 (Introduction), marked in magenta. Specifically, we now state that, in this paper, expressivity saturation denotes the observed reduction in realized affine-region usage under increasing task complexity. The term is therefore used in a descriptive sense, with respect to the number of affine regions actually realized by the trained network. We agree that understanding how this notion relates to broader optimization phenomena such as underfitting or implicit bias is an interesting direction, but this is beyond the scope of the present paper.
>
> 3. On the gap between the 1D theoretical guarantees and the higher-dimensional empirical observations.
>
> We have clarified this point in the revised manuscript by adding an explicit sentence in Section 1 (Introduction), in the paragraph “Our perspective,” marked in magenta. Specifically, we now state that the 1D theoretical results are probe-restricted and do not directly imply the higher-dimensional empirical observations, and that the two parts of the paper are intended to be complementary.
>
> 4. On comparisons or discussion relative to existing empirical works on region utilization.
>
> We have addressed this point in the revised manuscript by adding a dedicated discussion in the Appendix, in the section “Discussion,” marked in magenta. This added discussion compares our work with existing empirical and theory-grounded studies on region utilization, and clarifies how our contribution differs in terms of realized affine-region usage, exact bounded-domain enumeration, and the controlled task-complexity setting considered in this paper.
>
> 5. On practical implications.
>
> We have clarified this point in the revised manuscript by adding brief discussion of the practical implications and scope of our findings, marked in magenta. Our main practical message is interpretive rather than prescriptive: worst-case expressive capacity does not necessarily reflect the complexity actually realized after training. From this perspective, realized affine-region usage may be informative when comparing architectures or training settings in controlled regimes. At the same time, the manuscript now makes explicit that extending this perspective to broader architectures and real-world datasets remains an important direction for future work.

---

### Review · Reviewer_YZS6 · 2026-04-12

**Summary Of Contributions:**

This paper studies the gap between the theoretical affine-region capacity of piecewise-affine neural networks and the regions actually realized after training. On the theory side, it proves a rigorous architecture-dependent upper bound on the number of affine pieces a CPWA MLP can realize along any 1D probe, with a corollary giving a neuron-threshold lower bound for representing 1D targets of prescribed piece complexity. On the empirical side, it introduces exact 1D probe counting and exact bounded-domain enumeration in 2D and 3D, and uses these tools to document an "expressivity saturation" phenomenon: under fixed architectures and training protocols, increasing controlled task complexity (operationalized via random labels and sample-size sweeps) can sharply reduce realized region usage, often alongside degraded 2D decision boundaries. Optimizer/initialization ablations (Table 3) and a residual-connection appendix (Appendix C) support that the collapse is not tied to one configuration, though both still use the random-label task.

**Audience:**

Yes

**Audience Explanation:**

This paper would interest readers working on expressivity, neural network geometry, optimization, and theoretical deep learning, since it connects worst-case capacity questions to what trained networks actually realize using exact counting and clear visualizations rather than asymptotic arguments. The highlighted phenomenon — large theoretical capacity but reduced realized region usage under harder fitting regimes — is a useful perspective for both theoretical and empirical readers in TMLR's audience.

**Broader Impact Concerns:**

None. There is no broader impact concerns associated with this work.

**Claims And Evidence:**

Yes

**Claims Explanation:**

The central claims are well supported within the scope studied. Theorem 4 and Corollary 6 are proved rigorously in Appendix A, and the 1D/2D/3D counts are exact rather than estimated, with Table 2 showing region collapse by orders of magnitude (e.g., from ~3000 down to 35 at 10000 samples in late epochs) and Table 4 showing the same trend in 3D, while Tables 3 and 5 indicate the phenomenon is not a single-configuration artifact. That said, the evidence most directly supports a controlled statement about small fully connected ReLU MLPs with random labels as a proxy for high complexity, and the decision-boundary failure story is more correlational than mechanistic; Table 1 also shows that the theoretical lower bound is loose by orders of magnitude as a practical design tool (Q = 4 needs only 2 neurons by theory but 108 in practice, and Q = 24, 32 had no successful fits at all).

**Requested Changes:**

Overall the paper presents a sound storyline. Below are some suggested changes that could further strengthen the results:
* Narrow the wording of the main empirical claims so they match the actual scope of the evidence. Avoid phrasing that sounds universal over "task complexity" or CPWA networks in general, and instead emphasize that the phenomenon is established for low-dimensional fully connected ReLU MLPs with random-label/sample-size regimes as a proxy for high complexity.
* Strengthen statistical reporting in the 2D/3D experiments, since Tables 2, 4, and 5 appear to report single-run numbers with no variability information. Multiple seeds, error bars, and accompanying train accuracy/loss summaries would help readers separate optimization failure from geometric collapse, and the 3D evidence should be expanded beyond a single table at a single epoch with one architecture.
* Make explicit that the 1D theorem does not yet explain the higher-dimensional saturation phenomenon mechanistically. The current presentation places the rigorous probe-based bound and the higher-dimensional empirical collapse side by side without clarifying that the connection is motivational rather than causal.

---

> ### Author Response · Authors · 2026-04-19
> **Response to Reviewer YZS6**
>
> We thank the reviewer YZS6 for these helpful suggestions and have revised the manuscript accordingly.
>
> 1. Empirical claims are too broad and should be scoped to the actual experimental setting.
>
> We narrowed the wording of the main empirical claims to match the evidence more precisely. In the Introduction (“Our perspective”), we now state in purple that the phenomenon is observed in controlled low-dimensional fully-connected ReLU MLP settings under fixed architectures and training protocols, with random-label/sample-size regimes used as a proxy for increasing task complexity. We also revised, in magenta, the definition of “expressivity saturation” so that it is explicitly scoped to the controlled low-dimensional fully-connected ReLU MLP regimes studied here. In addition, Contribution 2 is revised in purple to reflect the same scope, and the Conclusion is revised in purple to restate the empirical findings in this qualified form. These changes are consistent with the existing Limitations section, which already clarifies that our empirical conclusions are confined to controlled low-dimensional settings and that random-label regimes are used only as a proxy rather than as a universal model of task complexity.
>
> 2. Strengthen statistical reporting in the 2D/3D experiments.
>
> We agreed that the original version did not make run-to-run variability sufficiently explicit in the 2D/3D experiments, and we revised the paper accordingly. In the updated manuscript, the main 2D and 3D results are now reported over 5 random seeds with all scalar summaries given as mean ± standard deviation. In particular, Table 2 now reports exact 2D bounded-domain affine-region counts across multiple training epochs rather than isolated single-run values, and Figure 3 now adds aggregated training summaries (epoch vs. training accuracy, epoch vs. training loss, epoch vs. exact region count, and region count vs. training accuracy) to help separate optimization behavior from geometric collapse. We made the same strengthening in 3D: Table 4 now reports exact counts for two architectures ([16,16,16] and [32,32,32]), across five sample sizes and multiple epochs (0, 10, 30, 50, 80, 100), all aggregated over 5 seeds, while Figure 5 adds the corresponding training-accuracy/loss/region-count summaries. We also added a 2D initialization/optimizer ablation in Table 3 and expanded the residual-connection appendix so that Table 5 now reports mean ± standard deviation over 5 seeds together with optimization summaries in Figure 9.
>
> 3. The connection between the 1D theorem and the higher-dimensional saturation phenomenon should be clarified as non-mechanistic.
>
> We now make this explicit in the Introduction (“Our perspective”), where we added in magenta that the 1D theoretical results are probe-restricted and do not directly imply the higher-dimensional empirical observations, and that the two parts of the paper are intended to be complementary. This is also consistent with the existing Limitations section, which already states that our theoretical results are probe-based and do not directly characterize full-dimensional region counts or off-probe geometry. We therefore clarify that the 1D theorem is not presented as a mechanistic explanation of the 2D/3D saturation phenomenon, but rather as a rigorous probe-level capacity result that complements the exact higher-dimensional empirical observations under controlled settings.

---

### Decision · Action_Editor_gYLf · 2026-06-05

**Recommendation:** Accept as is

**Audience:**

Yes

**Audience Explanation:**

The paper presents a technically sound and well-motivated study of realized expressivity in piecewise-affine neural networks. The work addresses an interesting and important question regarding the gap between worst-case expressive capacity and the complexity actually realized after training.

**Claims And Evidence:**

Yes

**Claims Explanation:**

The central claims are well supported -- the theoretical claims are rigorously proved in Appendix A, while experiments are in a simple controlled setting and support the paper's message. The reviewers asked for additional experiments and a more thorough comparison to the existing literature. All the reviewers are convinced by the rebuttal.